# Noise Contrastive Alignment of Language Models with Explicit Rewards

**Huayu Chen[1,2], Guande He[1,2], Lifan Yuan[1], Ganqu Cui[1], Hang Su[1,2,3], Jun Zhu[1,2]***
[1]Department of Computer Science and Technology, Tsinghua University
[2]Institute for AI, BNRist Center, Tsinghua-Bosch Joint ML Center, THBI Lab, Tsinghua University
[3]Zhongguancun Laboratory, Beijing, China

## Abstract

User intentions are typically formalized as evaluation rewards to be maximized when fine-tuning language models (LMs). Existing alignment methods, such as Direct Preference Optimization (DPO), are mainly tailored for pairwise preference data where rewards are implicitly defined rather than explicitly given. In this paper, we introduce a general framework for LM alignment, leveraging Noise Contrastive Estimation (NCE) to bridge the gap in handling reward datasets explicitly annotated with scalar evaluations. Our framework comprises two parallel algorithms, NCA and InfoNCA, both enabling the direct extraction of an LM policy from reward data as well as preference data. Notably, we show that the DPO loss is a special case of our proposed InfoNCA objective under pairwise preference settings, thereby integrating and extending current alignment theories. By comparing NCA and InfoNCA, we demonstrate that the well-observed decreasing-likelihood trend of DPO/InfoNCA is caused by their focus on adjusting relative likelihood across different responses. In contrast, NCA optimizes the absolute likelihood for each response, thereby effectively preventing the chosen likelihood from decreasing. We evaluate our methods in both reward and preference settings with Mistral-$8\times$7B and 7B models. Experiments suggest that InfoNCA/NCA surpasses various preference baselines when reward datasets are available. We also find NCA significantly outperforms DPO in complex reasoning tasks like math and coding. Code: `https://github.com/thu-ml/Noise-Contrastive-Alignment`.

## 1 Introduction

Aligning pretrained Language Models (LMs) with scalar rewards that reflect human intentions is crucial for enhancing their ability to follow instructions [35, 25]. These rewards can be given either explicitly or implicitly. Explicit rewards can be scalar ratings of human annotators or advanced models like GPT-4, while implicit rewards are usually preference labels assigned to pairwise responses.

One effective approach for aligning LMs with preference data is Direct Preference Optimization (DPO, [33]). DPO applies a reward training loss but parameterizes the reward model as the response likelihood ratio between two LMs, allowing for training reward models and extracting LM policies simultaneously. This approach is more streamlined and thus more favorable compared with traditional Reinforcement Learning (RL) methods [26], which typically require a two-stage training process: first training reward models, then extracting LM policies.

Despite its simplicity and effectiveness, DPO is only tailored for preference data ($x \to \{y_w > y_l\}$). When multiple responses are available, directly assigning a scalar reward to each response is usually more convenient and efficient than comparing them in a pairwise manner. The resulting reward

---

*The corresponding author

38th Conference on Neural Information Processing Systems (NeurIPS 2024).

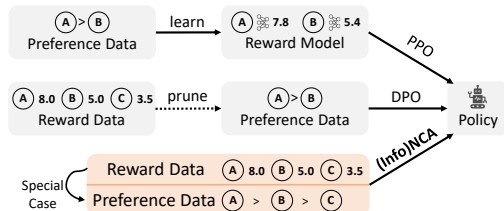

Figure 1: InfoNCA/NCA allows direct LM optimization for both reward and preference data.

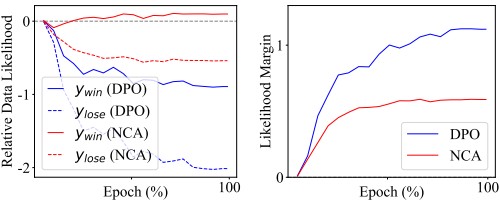

Figure 2: Pairwise NCA prevents chosen likelihood from decreasing while DPO cannot.

datasets ($x \to \{y_i, r_i\}_{1:K}$), however, cannot be directly leveraged for DPO training. Previous work [40] usually prunes reward datasets by selecting the best response and pairing it with a random remaining one. This is suboptimal as all reward values and additional dispreferred responses are thrown away in its data-preprocessing process.

To address this issue, we present **InfoNCA**, an alignment method that allows directly extracting LM policies from both reward datasets and preference datasets with arbitrary response numbers (Figure 1). Notably, InfoNCA subsumes DPO loss as a special case under pairwise preference settings and can thus be seen as a natural extension of DPO (Sec. 3.2). With strong theoretical guarantees, we show DPO is a binary classification loss while InfoNCA is its multi-category version (Figure 3). However, unlike DPO which is built upon assumptions of Bradley-Terry models or Plackett-Luce models, InfoNCA is strictly derived from Information Noise Contrastive Estimation (InfoNCE, [24]), an established contrastive method that is widely applied in language and visual representation learning [31]. This closes the theoretical gap between current preference alignment methods and classic contrastive learning frameworks.

A well-observed problem with DPO is that the likelihood of the preferred response tends to *decrease* throughout training [27, 32]. Similar trends are shared by InfoNCA. We find this issue arises mainly from InfoNCA/DPO's focus on adjusting the *relative* likelihood across different responses per instruction. In Sec. 4, we propose **NCA** as an alternative alignment method to InfoNCA to mitigate this problem. NCA differs from InfoNCA by only loss definition and is also suitable for both preference and reward datasets. However, NCA is built on NCE [14], a parallel contrastive learning algorithm to InfoNCE, which optimizes the *absolute* data likelihood during training. In practice, NCA effectively prevents the chosen likelihood from decreasing (Figure 2).

We evaluate our methods on Mistral-7B and $8\times$7B models from two dimensions. When reward datasets [9] are available, we show that directly applying our reward-based alignment offers clear improvement compared with preference-based algorithms, achieving higher evaluation rewards in GPT-4 [47, 19] evaluations. We further validate this improvement comes from InfoNCA/NCA's ability to fully leverage the additional suboptimal responses. When only preference data is given [44], we compare pairwise NCA against the DPO loss. Our experimental results spanning various benchmarks show that NCA outperforms DPO in complex reasoning tasks such as math and coding.

Our main contributions: 1. We bridge the theoretical gap between DPO and classic contrastive learning theories. InfoNCA and NCA are uniquely suited for both reward and preference data, offering a general framework that integrates preference-based algorithms. 2. We show that suboptimal responses are also important for LM optimization. Our method outperforms various preference methods by fully exploiting data information in reward datasets. 3. NCA effectively mitigates the data likelihood decline issue of DPO and offers practical performance improvement.

## 2 Background: Direct Preference Optimization

LM alignment is essentially a constrained policy optimization problem:

$$\max_{\pi_\theta} \mathbb{E}_{p(x)}\left[\mathbb{E}_{\pi_\theta(y|x)} r(x, y) - \alpha D_{\mathrm{KL}}\left(\pi_\theta(\cdot|x) || \mu(\cdot|x)\right)\right], \tag{1}$$

where $\mu$ represents the pretrained LM. $x$ and $y$ are respectively instructions and responses. $r$ is a reward function that reflects human intentions. $\alpha$ is some temperature coefficient. Prior work [30, 29]

has proved that the optimal solution for the optimization problem in Eq. 1 is:

$$\pi^*(y|x) = \mu(y|x)\frac{e^{r(x,y)/\alpha}}{Z(x)} \propto \mu(y|x)e^{r(x,y)/\alpha}. \tag{2}$$

Direct Preference Optimization (DPO) [33] assumes we only have access to some pairwise preference data $x \rightarrow \{y_w > y_l\}$ for each instruction $x$. The preference probability of human annotators is modeled by a learnable implicit reward model $r_\theta$ under Bradley-Terry theories [2]:

$$\pi_\theta(y_w > y_l|x) := \sigma(r_\theta(y_w, x) - r_\theta(y_l, x)),$$

where $\sigma$ is the sigmoid function. To learn $r_\theta$, DPO simply adopts a binary classification loss:

$$\mathcal{L}_{\text{DPO}} = -\mathbb{E}_{\{x,y_w>y_l\}} \log \sigma(r_\theta(y_w, x) - r_\theta(y_l, x)). \tag{3}$$

In practice, the latent function $r_\theta$ is parameterized by the log-likelihood ratio between $\pi_\theta$ and $\mu$:

$$r_\theta(x, y) := \beta \log \frac{\pi_\theta(y|x)}{\mu(y|x)},$$

where $\beta$ a linear coefficient for scaling $r_\theta$. This parameterization is crucial because it ensures $\pi^\theta(y|x) \propto \mu(y|x)e^{r_\theta(x,y)/\beta}$ constantly hold. It transforms generative policy optimization into a simple discriminative classification task: When $r_\theta = r$ and $\beta = \alpha$ are satisfied, we naturally have $\pi_\theta = \pi^*$.

## 3   InfoNCA: Extending DPO from Preference to Explicit Rewards

Compared with constructing preference datasets, annotating each response with scalar rewards can be more flexible and convenient. Preference methods are only suitable for pairwise data ($x \rightarrow \{y_w > y_l\}$) and would require $C_K^2$ evaluations for comparing $K$ responses. In contrast, reward datasets ($x \rightarrow \{y_i, r_i\}_{1:K}$) allow an arbitrary number of responses per prompt with $K$ evaluations.

Despite its simplicity in handling preference data, DPO is not tailored for reward datasets. We introduce a new alignment method termed InfoNCA to mitigate this gap. We first strictly derive InfoNCA in Sec. 3.1. We show that reward alignment can be solved by constructing a classification problem to identify the optimal response from multiple candidates. We then demonstrate that InfoNCA subsumes DPO as a special case and thus is a natural extension of DPO (Sec. 3.2).

### 3.1   Reward Alignment through Multi-Class Classification

In essence, DPO represents response rewards as LM likelihoods and constructs a *binary* classification task for learning the reward model. Given that there are more than two ($K > 2$) responses per prompt in reward datasets, we seek to construct a *multi-class* classification task for learning reward models from explicit rewards instead of preference labels. We begin by formally defining this task:

Consider a batch of $K$ responses $\{y_i\}_{1:K}$ for an instruction $x$. $\{y_i\}_{1:K}$ consists of one optimal response $y_\nu$ that is sampled from $\pi^*(y|x) \propto \mu(y|x)e^{r(x,y)/\alpha}$, and $K - 1$ suboptimal noises independently sampled from $\mu(y|x)$. $\nu \in 1 : K$ is the random index of that optimal response. Our goal is to identify which of the $K$ candidates is $y_\nu$, given only reward labels $r(y_i)$ for each response.

Intuitively, the response with higher rewards should have a higher probability of being the target response. This intuition can be more rigorously expressed:

**Proposition 3.1** (proof in Appendix A.1). *Given the above $K$ response candidates and their respective rewards, the posterior probability for the $\nu$-th response being drawn from $\pi^*$ is*

$$p(\nu|x, \{y_i\}_{1:K}) = \frac{e^{r(x,y_\nu)}}{\sum_{i=1}^{K} e^{r(x,y_i)}}. \tag{4}$$

This finding is highly appealing because it shows response optimality is solely related to response rewards. This allows us to conveniently train reward models $r_\theta$ via maximum likelihood (MLE).

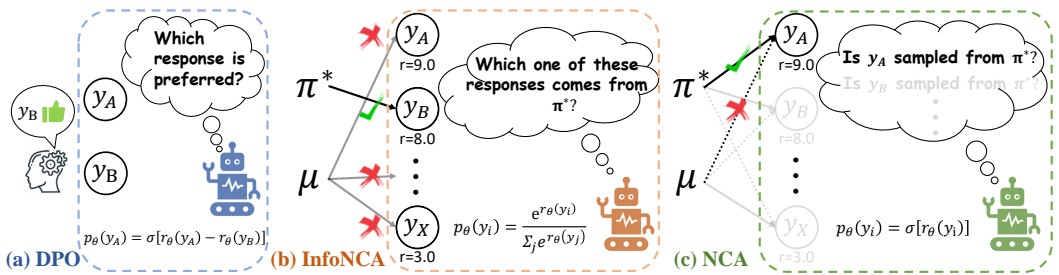

Figure 3: DPO, InfoNCA, and NCA all optimize LLM through classification tasks. **DPO** compares two responses and tells which one is preferred. **InfoNCA** compares multiple responses and identifies the one sampled from $\pi^*$ (Sec. 3.1). **NCA** predicts the model source of a single response (Sec. 4.1).

**Theorem 3.2** (InfoNCA, proof in Appendix A.1). *We define $\pi^*(y|x) \propto \mu(y|x)e^{r(x,y)/\alpha}$ and $\pi_\theta(y|x) \propto \mu(y|x)e^{r_\theta(x,y)}$. For any $K > 1$, $\alpha > 0$, we have:*

*(a) Equivalent objective. The MLE objective for training $r_\theta$ has an equivalent form:*

$$\max_\theta \mathbb{E}_{p(x,\{y_i\})} \log p_\theta(\nu|x, \{y_i\}) \Leftrightarrow \min_\theta -\mathbb{E}_{p(x)} \prod \mu(y_i|x) \sum_{i=1}^{K} \left[ \frac{e^{r(x,y_i)/\alpha}}{Z(x)} \log \frac{e^{r_\theta(x,y_i)}}{\sum_{j=1}^{K} e^{r_\theta(x,y_j)}} \right], \quad (5)$$

*where $Z(x) = \mathbb{E}_{\mu(y|x)} e^{r(x,y)/\alpha}$.*

*(b) Optimal solution. Assume unlimited model capacity. The optimal $r_{\theta^*}$ and $\pi_{\theta^*}$ for solving (5) are*

$$r_{\theta^*}(x,y) = r(x,y)/\alpha + C(x),$$
$$and \quad \pi_{\theta^*}(y|x) = \pi^*(y|x) \propto \mu(y|x)e^{r(x,y)/\alpha},$$

*where $C(x)$ is an arbitrary function conditioning on $x$.*

In practical implementation of Eq. 5, we estimate $Z(x) \approx \frac{1}{K} \sum e^{r_j/\alpha}$ and parameterize $r_\theta(x,y) := \beta \log \frac{\pi_\theta(y|x)}{\mu(y|x)}$ following DPO (Eq. 2). The loss function becomes

$$\mathcal{L}_\theta^{\text{InfoNCA}}\big(x, \{y_i, r_i\}_{1:K}\big) = -\sum_{i=1}^{K} \Big[ \underbrace{\frac{e^{r_i/\alpha}}{\sum_{j=1}^{K} e^{r_j/\alpha}}}_{\text{soft labels}} \log \underbrace{\overbrace{e^{r_\theta(x,y_i)}}^{\text{model logits}}}_{\text{predicted probability}} \Big]_{r_\theta(x,y):=\beta \log \frac{\pi_\theta(y|x)}{\mu(y|x)}}. \quad (6)$$

The loss function in Eq. 6 is termed **InfoNCA**, where **A** stands for **A**lignment. This naming reflects its functional similarity to the Infomation Noise Contrastive Estimation (InfoNCE, [24]). Both methods transform generative modeling problems into classification tasks by contrasting multiple data points.

**How does InfoNCA work?** InfoNCA loss (Eq. 6) can be seen as a $K$-category cross-entropy loss. The soft label is calculated by dataset rewards through a softmax operation. The model's predictions are represented by learned reward $r_\theta$. The loss reaches 0 when $r_{\theta^*}(x,y) = r(x,y)/\alpha + C(x)$.

### 3.2 InfoNCA Subsumes DPO as A Special Case

Below we show that DPO is a special case of InfoNCA asymptotically. Specifically, setting response number $K = 2$ and reward temperature $\alpha \to 0$, we can fully recover the DPO objective:

$$\mathcal{L}_\theta^{\text{InfoNCA}}\big(x, \{y_i, r_i\}_{1:K}\big) = -\sum_{i=1}^{K} \Big[ \mathbb{1}(r_i = r_{\max}) \log \frac{e^{r_\theta(x,y_i)}}{\sum_{j=1}^{K} e^{r_\theta(x,y_j)}} \Big] \qquad (\alpha \to 0)$$

$$= -\log \frac{e^{r_\theta(x,y_w)}}{e^{r_\theta(x,y_w)} + e^{r_\theta(x,y_l)}} \qquad (K = 2, \text{ suppose } r_w > r_l)$$

$$= -\log \sigma(r_\theta(x,y_w) - r_\theta(x,y_l)) \qquad (\text{DPO loss, Eq. 3})$$

| Alignment Method | InfoNCA (Sec. 3) | | NCA (Sec. 4) |
|---|---|---|---|
| Modeling Target | $\pi^*(y\|x) \propto \mu(y\|x)e^{r(x,y)/\alpha}$ | | |
| Model Definition | $\pi_\theta(y\|x) \propto \mu(y\|x)e^{r_\theta(x,y)}$ | | $\pi_\theta(y\|x) = \mu(y\|x)e^{r_\theta(x,y)}$ |
| Reward Dataset | $x \to \{y_i, r_i\}_{1:K}$ | | |
| Loss ($K>1, \alpha>0$) | $-\sum_{i=1}^K \left[ \frac{e^{r_i/\alpha}}{\sum_j e^{r_j/\alpha}} \log \frac{e^{r_\theta(x,y_i)}}{\sum_j e^{r_\theta(x,y_j)}} \right]$ | $-\sum_{i=1}^K \left[ \frac{e^{r_i/\alpha}}{\sum_j e^{r_j/\alpha}} \log \sigma(r_\theta(x,y_i)) + \frac{1}{K} \log \sigma(-r_\theta(x,y_i)) \right]$ | |
| Preference Dataset | $x \to \{y_w > y_l\}$ | | |
| Loss ($K=2, \alpha\to 0$) | $-\log \sigma(r_\theta(x,y_w) - r_\theta(x,y_l))$ (DPO) | $-\log \sigma(r_\theta(x,y_w)) - \frac{1}{2} \sum_{y \in \{y_w, y_l\}} \log \sigma(-r_\theta(x,y))$ | |
| Loss Type | InfoNCE loss [24] | | NCE loss [14] |
| Optimizing Target | *relative* value of log likelihood ratio | | *absolute* value of log likelihood ratio |
| Optimal $r_{\theta*}(x,y)$ | $r(x,y)/\alpha + C(x)$ | | $r(x,y)/\alpha - \log \mathbb{E}_{\mu(y\|x)} e^{r(x,y)/\alpha}$ |
| $r_{\theta*}(x, y_{\text{best}}) \geq 0$ ? | not guaranteed | | ✔ |

Table 1: Comparison of NCA and InfoNCA algorithm for aligning language models. Both reward loss and pairwise preference loss are given. We provide pseudocode in Appendix B.

**Empirical effect for varying hyperparameter $K$ and $\alpha$.** As indicated by the derivation above, the root difference between preference-based and reward-based methods lies in the choices of $K$ and $\alpha$.

$K$ affects how accurately we can estimate the partition function $Z(x) \approx \sum_{j=1}^K e^{r_j/\alpha}$ in Eq. 5. In practice, we find larger $K$ can lead to better performance (Sec. 5.1). $\alpha$ indicates a trade-off between diversity and optimality. At $a \to 0$, the InfoNCA loss increases the likelihood only for the optimal response and decreases it for all other responses, turning the reward dataset $x \to \{y_i, r_i\}$ into a preference dataset $x \to \{y_w > y_l\}$. We provide ablation studies of $\alpha$ and $\beta$ in Appendix D.

## 4 NCA: Fixing Decreased Response Likelihood Issue for InfoNCA

A well-observed issue with DPO is that the likelihood of *all* responses continually *decrease* throughout training [27, 32]. We find InfoNCA shares this trend due to their inherent equivalence. Decreased data likelihood is concerning because it directly contradicts the maximum likelihood objective for supervised training and may eventually harm performance [44].

We hypothesize the main cause of this decreasing likelihood is that InfoNCA methods only adjust *relative* rewards among responses, rather than optimizing their *absolute* value. To address this problem, we take inspiration from NCE, another contrastive learning method parallel to InfoNCE, and propose **NCA**(lignment) (Sec. 4.1). Similar to InfoNCA, NCA can also guarantee convergence to the optimal LM policy under ideal conditions (Theorem 4.2). However, it directly learns the absolute reward for each response, thereby counteracting the decreasing likelihood trend (Sec. 4.2).

### 4.1 Reward Alignment through Absolute Reward Prediction

To avoid optimizing relative rewards across multiple responses, we construct a binary classification task that deals with a single response.

Specifically, imagine sampling a response $y$ randomly from either the optimal LM $\pi^*(y|x) = \mu(y|x)\frac{e^{r(x,y)/\alpha}}{Z(x)}$, or the pretrained LM $\mu(y|x)$. The marginal probability of $y$ is $p(y|x) := \frac{1}{2}\mu(y|x) + \frac{1}{2}\pi^*(y|x)$. Our goal is to guess its model source when given a response $y$ and its reward $r(y)$.

**Proposition 4.1** (proof in Appendix A.2). *Let a binary variable $\nu = 1$ indicates the response $y$ is sampled from $\pi^*$. The posterior probability of the distribution source given the response $y$ satisfies:*

$$p(\nu = 1|x, y) = \frac{\pi^*(y|x)}{\mu(y|x) + \pi^*(y|x)} = \frac{e^{r(x,y)/\alpha}}{Z(x) + e^{r(x,y)/\alpha}}. \tag{7}$$

Note that $p(\nu|x, y)$ is related to the partition function $Z(x)$. In order to represent model likelihood $p_\theta(\nu|x, y)$ by only employing $r_\theta$ similarly to Proposition 3.1, we have to redefine

$\pi_\theta(y|x) = \mu(y|x)e^{r_\theta(x,y)}$ by absorbing $Z_\theta$ into $r_\theta$. Then we have

$$p_\theta(\nu = 1|x,y) = \frac{\pi_\theta(y|x)}{\mu(y|x) + \pi_\theta(y|x)} = \sigma(r_\theta(x,y)). \tag{8}$$

Similarly to Theorem 3.2, we can derive a MLE-based training objective for optimizing $r_\theta$.

**Theorem 4.2** (NCA, proof in Appendix A.2). *Let $\alpha > 0$, we have the maximum likelihood objective:*

*(a) Equivalent objective.*

$$\max_\theta \mathbb{E}_{p(x,y)} \log p_\theta(\nu|x,y) \Leftrightarrow \min_\theta -\mathbb{E}_{p(x)\mu(y|x)}\Big[\frac{e^{r(x,y)/\alpha}}{Z(x)}\log\sigma(r_\theta(x,y)) + \log\sigma(-r_\theta(x,y))\Big], \tag{9}$$

*where $Z(x) = \mathbb{E}_{\mu(y|x)}e^{r(x,y)/\alpha}$.*

*(b) Optimal solution. Assume unlimited model capacity. The optimal $r_{\theta^*}$ and $\pi_{\theta^*}$ for solving (9) are*

$$r_{\theta^*}(x,y) = r(x,y)/\alpha - \log\mathbb{E}_{\mu(y|x)}e^{r(x,y)/\alpha}, \tag{10}$$
$$and \quad \pi_{\theta^*}(y|x) \propto \mu(y|x)e^{r(x,y)/\alpha}.$$

For reward datasets $(x \to \{y_i, r_i\}_{1:K})$, we estimate $Z(x) \approx \sum_{i=1}^{K} e^{r_i/\alpha}$ in Eq. 9 and construct $r_\theta$ similarly to InfoNCA:

$$\mathcal{L}_\theta^{\text{NCA}}(x, \{y_i, r_i\}_{1:K}) = -\sum_{i=1}^{K}\Big[\underbrace{\frac{e^{r_i/\alpha}}{\sum_{j=1}^{K}e^{r_j/\alpha}}}_{\substack{\text{softmax weight}}}\underbrace{\log\sigma(r_\theta(x,y_i))}_{\substack{\text{optimizer}\uparrow\\ \text{(increasing force)}}} + \frac{1}{K}\underbrace{\log\sigma(-r_\theta(x,y_i))}_{\substack{\text{regularizer}\downarrow\\ \text{(decreasing force)}}}\Big]_{r_\theta(x,y):=\beta\log\frac{\pi_\theta(y|x)}{\mu(y|x)}} \tag{11}$$

**How does NCA work?** The loss function for NCA involves two opposing forces that jointly determine the trend of increasing or decreasing $r_\theta(x,y)$. Since $\log\sigma(\cdot)$ is a monotonically increasing function, the first term in Eq. 11 tends to increase $r_\theta(x,y)$ while the second term tends to decrease it.

At the start of training, when $r_\theta = -r_\theta = 0$, the direction of the combined force for $r_\theta$ is decided by the difference in their weights, expressed as $\frac{e^{r_i/\alpha}}{\sum_{j=1}^{K}e^{r_j/\alpha}} - \frac{1}{K}$. Responses with higher rewards would, in principle, attain higher likelihood after training.

### 4.2 Connection between NCA and InfoNCA/DPO

Although both NCA and InfoNCA originate from solving a noise contrastive classification problem, their optimization targets are markedly different (Table 1).

InfoNCA and DPO both calibrate *relative* values of reward models across various responses $\{y_i\}_{1:K}$ for an instruction $x$. In other words, the absolute value of $r_\theta(x,y)$ is not directly constrained. This can lead to some counterintuitive behaviors. For instance, the learned reward for even the highest-reward response could decrease over time without contradicting the loss definition, as long as the reward margin keeps increasing. This could lead to poor performance or training instability (Sec. 5.2).

In contrast, NCA specifically focuses on optimizing *absolute* values of the reward model. This characteristic is determined by its model definition: $\pi_\theta(y|x) = \mu(y|x)e^{r_\theta(x,y)}$, where $r_\theta$ has to be *self-normalized*: $\mathbb{E}_{\mu(y|x)}e^{r_\theta(x,y)} = 1$. In practice, NCA effectively prevents the likelihood of the preferred responses from decreasing. We find this is particularly helpful for math and coding tasks.

## 5 Experiments

We mainly seek to answer two questions in our experiments:

1. If we have access to reward-annotated datasets with >2 responses per prompt, does InfoNCA or NCA offer empirical improvement compared with preference-based approaches that simply prune reward datasets into preference datasets? (Sec. 5.1)
2. If only pairwise preference data is available, when should one choose NCA over DPO? What benefits does NCA offer? (Sec. 5.2) Note that InfoNCA is exactly DPO in this setting.

| | Name | Annotation Type | MT-bench | AlpacaEval | Win vs. DPO |
|---|---|---|---|---|---|
| **Baseline** | Mixtral-7B-sft | SFT Data | 6.45 | 85.20 | - |
| | +KTO [11] | Preference | 7.12 | 91.93 | - |
| | +IPO [1] | Preference | 7.45 | 90.62 | - |
| | +DPO (Zephyr-$\beta$) | Preference | 7.34 | 90.60 | 50.0 |
| | +DPO×3 | Preference | 7.22 | 91.60 | 58.1 |
| | +DPO×$C_4^2$ | Preference | 7.38 | 90.29 | 48.1 |
| **Ours** | **+InfoNCA** | Reward | **7.63** | **92.35** | 56.9 |
| | **+NCA** | Reward | 7.52 | 90.31 | **59.4** |
| **Reference** | Mixtral-ORPO-$\beta$ | Preference+SFT | 7.32 | 91.41 | - |
| | Mistral-7B-instruct | SFT Data | 6.84 | 92.78 | - |
| | LLaMA2-chat-70b | Reward Model | 6.86 | 92.66 | - |
| | GPT-4 | Reward Model | 9.18 | 93.78 | - |

Table 2: Comparison between reward-based methods (InfoNCA, NCA) and preference-based methods (DPO, IPO, etc.) in LLM alignment. We focus on the general instruction-following abilities of each method measured by GPT-4 evaluations and human preference. The highest number in each benchmark is **bolded** and the second highest is underlined.

## 5.1 Aligning Language Models with Explicit Rewards

**Reward dataset and Evaluation metric.** We consider `UltraFeedback` [9], an instruction-following dataset annotated by GPT-4. This dataset comprises ∼64k instructions. Each instruction has 4 responses generated by various LMs. GPT-4 rates each response with a scalar reward on a scale of 0-10. Prior research indicates that these GPT-4 rewards closely align with human annotations [47], establishing them as an efficient, cost-effective alternative to human feedback. In order to align exactly with the definition of dataset rewards, we similarly choose well-acknowledged GPT4-based benchmarks like MT-bench [47] and AlpacaEval [19] for evaluation. Human preference studies are also conducted on evaluation prompts from MT-bench. The rating system is in Appendix C.

**InfoNCA and NCA outperform preference-based methods given reward dataset.** To handle reward datasets with $K > 2$ responses per instruction, one approach is to simply prune them into pairwise data and apply preference learning like DPO. For instance, Zephyr [40] selects the highest-reward response and a random remaining one from UltraFeedback for each instruction. This procedure discards two additional suboptimal responses in the dataset as well as their reward information.

In Table 2, we fine-tune a `Mistral-7B` model on UltraFeedback and compare InfoNCA/NCA against the DPO baseline. Results show that our methods outperform preference baselines. This improvement can be attributed to InfoNCA/NCA's ability to exploit all information in the reward dataset.

| Method | K=2 | K=3 | K=4 |
|---|---|---|---|
| InfoNCA (MT-bench) | 73.8 | 75.9 | 76.3 |
| InfoNCA (Alpaca) | 90.7 | 90.2 | 92.4 |
| NCA (MT-bench) | 73.2 | 73.3 | 75.2 |
| NCA (Alpaca) | 89.9 | 90.3 | 90.3 |
| **Average** | **81.9** | **82.4** | **83.5** |

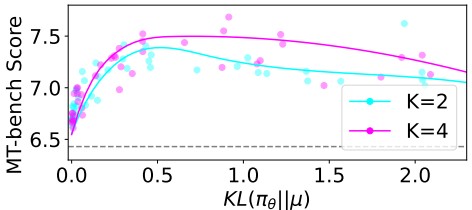

Figure 4: More suboptimal responses can also increase LLM's instruction-following ability. We fix the highest-reward response in the UltraFeedback dataset and ablate the number of suboptimal responses per prompt, resulting in different contrastive response numbers $K$ during training. **Left:** Evaluation results under the same set of hyperparameters. **Right:** Performance-KL trade-off under various $\alpha$ and $\beta$. Each dot represents an independent experiment trained for 1 epoch.

**Suboptimal responses are also important.** Previous practices always ensure selecting the highest-performing response when constructing preference data. The assumption behind this strategy is that the dataset's best-performing response determines the upper limit of alignment performance. However, our experiments contradict this assumption. Results in Figure 4 indicate that extra suboptimal responses can also be advantageous for policy training. Specifically, we observe consistent

| Model | Reasoning BBH (CoT) | Coding | | Math | | | | | Avg. |
| | | LeetCode | HumanEval | GSMPLUS | MATH | TheoremQA | SVAMP | ASDiv | |
|---|---|---|---|---|---|---|---|---|---|
| Mixtral-7B-SFT | 60.9 | 3.3 | 28.1 | 28.5 | 5.8 | 7.0 | 26.9 | 35.8 | 24.5 |
| + DPO | **61.7** | 2.2 ↓ | **31.7** | 12.1 ↓ | 6.4 | 9.8 | 34.1 | 46.1 | 25.5 |
| + NCA | 60.8 ↓ | **3.3** | 26.8 ↓ | **32.3** | **11.7** | **11.0** | **65.3** | **74.3** | **35.7** |
| Mixtral-8×7B-SFT | **75.6** | 16.7 | 61.0 | 57.6 | 40.1 | 25.9 | 85.9 | 87.5 | 56.3 |
| + DPO | 74.9 ↓ | 17.2 | 47.6 ↓ | 55.8 ↓ | 35.3 ↓ | **26.9** | 67.3 ↓ | 75.7 ↓ | 50.1↓ |
| + NCA | **75.6** | **21.1** | **62.8** | **61.5** | **41.6** | 26.9 | **86.8** | **86.9** | **57.9** |

Table 3: Alignment results for UltraInteract. We mark numbers that have decreased (↓) after training.

performance improvements when increasing the number of data responses from $K = 2$ to $K = 4$ for both InfoNCA and NCA algorithms, across various hyperparameters.

**Combinatorial DPO are suboptimal solutions.** Regarding the performance improvement offered by more suboptimal responses, one might predict that applying the DPO to a combinatorially constructed preference dataset would yield results comparable to NCA/InfoNCA. To investigate this, we examined two variants of DPO that utilize all available responses in UltraFeedback.

**DPO×3:** We pair the highest-performing response with each of the remaining three separately.

**DPO×$C_4^2$:** We sum up all DPO loss possibilities for two out of the four responses.

Our experiments, detailed in Table 2, reveal that naively applying combinatorial DPO loss to leverage all response information underperforms InfoNCA/NCA. The DPO×3 shows some benefit, while DPO×$C_4^2$ is harmful compared with simple data pruning. This is expected because InfoNCA and NCA possess theoretical guarantees (Theorem 3.2 and Theorem 4.2) that ensure convergence to the optimal LM policy whereas combinatorial preference methods do not.

## 5.2 NCA vs. DPO in Aligning Language Models with Pairwise Preference

In previous experiments, our focus is on the reward dataset with $K > 2$ responses per prompt ($x \rightarrow \{y_i, r_i\}_{1:K}$). However, at present most alignment datasets are pairwise ($x \rightarrow \{y_w > y_l\}$), making it essential also to evaluate our proposed methods in pairwise preference settings.

Since InfoNCA is equivalent to DPO when only pairwise preference data is available (Sec. 3.2), we will focus on comparing and clarifying the differences between the DPO and NCA algorithms.

**Preferecne dataset and evaluation metrics.** We consider fine-tuning Mistral-7B and Mistral-8×7B models on UltraInteract [44], a pairwise alignment dataset specifically designed for complex reasoning tasks. Before alignment, we perform SFT on UltraInteract's preferred responses for the 8×7B model and use the existing Mistral-SFT model in Sec. 5.1. We evaluate the model's performance in various challenging tasks. This includes BBH-Hard [39] for CoT reasoning, HumanEval [4] and LeetCode [12] for coding, GSM-Plus [18], MATH, TheoremQA [6], SVAMP [28], and ASDiv [22] for math.

**DPO may hurt reasoning performance while NCA helps.** Results are presented in Table 3. Overall, NCA consistently outperforms DPO in various benchmarks. Notably, we observe DPO hurts the overall performance in most reasoning tasks regarding the Mixtral-8×7B-SFT model. This indicates that DPO might not be suitable for improving reasoning abilities, which echoes findings in concurrent work [44]. In contrast, NCA shows clear improvement on both the 7B and 8×7B models.

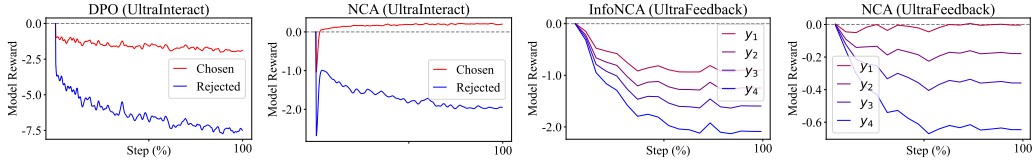

Figure 5: Comparision of data likelihood between InfoNCA/DPO and NCA.

**NCA prevents the chosen-likelihood from decreasing.** What distinct optimization characteristics could cause performance differences between pairwise NCA and DPO? To understand this, we empirically inspect how the data likelihood changes during training. As shown in Figure 5. The likelihood of preferred responses interestingly decreases after DPO training and increases for NCA training. This pattern is consistent across both preference and reward learning. The decreasing chosen-likelihood trend is concerning because it directly contradicts the maximum-likelihood objective used

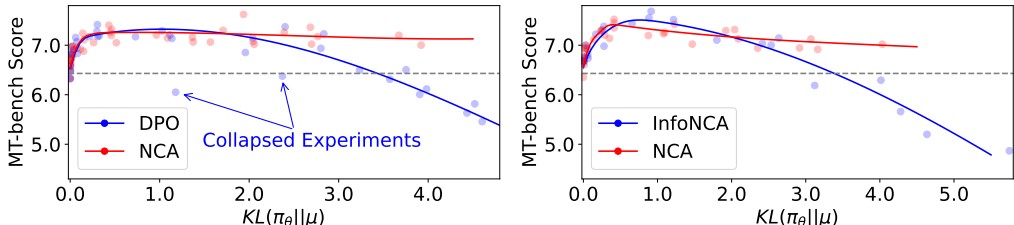

Figure 6: NCA is more robust to hyperparameter changes and KL derivations. **Left:** Ablation results of $\alpha$ and $\beta$ for UltraFeedback-binarized. **Right:** Results for UltraFeedback-reward ($K = 4$).

during the SFT stage. This drawback is exacerbated in reasoning tasks, where the preferred response is often the ground truth answer. Consequently, we hypothesize that NCA's superior performance in reasoning tasks is due to its ability to avoid decreasing chosen likelihood.

Since DPO is essentially a specialization of InfoNCA, their contrasting likelihood trends can be explained theoretically. As we have elaborated in Sec. 4.2, NCA adjusts the absolute likelihood of data, while DPO/InfoNCA only considers relative likelihood across different responses. Thus, a declining chosen likelihood directly contradicts NCA's training objective but not DPO's.

**Empirical takeaway: When to choose NCA over DPO?** DPO and pairwise NCA have similar theoretical guarantees. Their different performance in alignment tasks is largely empirical, depending on the specific characteristics of datasets and the nature of tasks. Our observations show that NCA is more suitable for reasoning tasks such as math and coding (Table 3), where high-quality responses are sparse, and adhering closely to the preferred responses in the dataset is critical. DPO may be more suitable for general instruction-following tasks like summarization/role-playing (Table 2), where datasets only reflect human relative preference but do not contain "golden" answers. In essence, NCA benefits from better dataset regularization, while DPO relies more on LLMs' generalization abilities.

We also observe that NCA has a greater tolerance for divergence from the initial SFT policy and is more robust to hyperparameter changes. As evidenced in Figure 6, we conduct a grid search on $\alpha$ and $\beta$. DPO can drastically fail to improve alignment performance if the learned policy strays too far from the SFT policy, and may randomly collapse under certain hyperparameters. In contrast, the NCA method does not exhibit similar issues. These observations suggest that NCA may be a worthwhile alternative if DPO training is unstable.

## 6 Related Work

**Language model alignment.** Current approaches cater to either explicit reward data or preference data, often lacking the versatility to address both concurrently. Reinforcement Learning [34] is inherently suitable for explicit reward scenarios. However, its on-policy nature necessitates learning a reward model from data first, leading to an indirect two-stage optimization process [8, 26, 36]. Recent developments in preference-based alignment techniques [33, 1, 11, 42, 16, 13] have streamlined this process. They enable direct alignment of LMs through a singular loss, but this comes at the expense of being confined to pairwise preference data. Other alignment approaches [43, 37, 45, 38] are also not tailored for aligning with reward datasets. Recent work [3] attempts to extend DPO's parameterization technique to explicit reward contexts. However, it only considers binary rewards. In comparison, our methods can handle both continuous rewards and preference data.

**Noise contrastive estimation.** NCE [14] and its variant, InfoNCE [24], are established optimization methods for training unnormalized generative models [21]. NCE primarily leverages a binary classification loss and can be applied in self-supervised representation learning. Examples are Word2Vec [23], MoCo [15], and SimCLR [5]. InfoNCE is related to maximizing mutual information between two distributions through a multiclass cross-entropy loss. It has successful applications in representation learning, such as CLIP [31]. It is also widely used in language modeling [7], diffusion modeling [20], and reinforcement learning [17].

# 7 Conclusion

In this work, we formally consider the language model alignment problem in the context of explicit reward settings. By adeptly harnessing the NCE and InfoNCE theories, we introduce two practical algorithms: NCA and InfoNCA. Our proposed methods are uniquely suited for both reward data and preference data, including DPO as a special case. Our experiments show that reward-based alignment methods outperform preference baseline by fully leveraging suboptimal responses in reward datasets. In preference settings, pairwise NCA outperforms DPO in complex reasoning tasks by effectively preventing data likelihood from decreasing.

## Acknowledgments and Disclosure of Funding

We especially thank Cheng Lu, who greatly inspires us in linking the NCA algorithm with NCE theories. We also thank Jiuhai Chen, and Tianlin Liu for their suggestions on Zephyr result reproduction. We thank Bingrui Li, and Weiyu Huang for their help with the experimental setup. We thank Github user Wing Lian for integrating the NCA algorithm into the trl library. We thank Haosheng Zou for providing feedback on our method.

This work was supported by NSFC Projects (Nos. 62350080, 92370124, 92248303, 62276149, 62061136001, 62076147), BNRist (BNR2022RC01006), Tsinghua Institute for Guo Qiang, and the High Performance Computing Center, Tsinghua University. J. Zhu was also supported by the XPlorer Prize.

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

# A Proof of Theorems

## A.1 InfoNCA Objective

Recall that our optimal language policy is

$$\pi^*(y|x) = \mu(y|x)\frac{e^{r(x,y)/\alpha}}{Z(x)}. \tag{Eq. 2}$$

Consider a batch of $K$ responses $\{y_i\}_{1:K}$ for an instruction $x$. $\{y_i\}_{1:K}$ consists of one optimal response $y_\nu$ that is sampled from $\pi^*(y|x) \propto \mu(y|x)e^{r(x,y)/\alpha}$, and $K-1$ suboptimal noises independently sampled from $\mu(y|x)$. $\nu \in 1:K$ is the random index of that optimal response. The the joint probability for $\{y_i\}_{1:K}$ is

$$p^{\text{joint}}(\{y_i\}_{1:K}|x,\nu) = \pi^*(y_\nu|x)\prod_{i\neq\nu}\mu(y_i|x) = \frac{\pi^*(y_\nu|x)}{\mu(y_\nu|x)}\prod_{i=1}^{K}\mu(y_i|x).$$

Given that the prior satisfies $p(\nu=1) = p(\nu=2) = ... = p(\nu=K) = \frac{1}{K}$, the data posterior is

$$p^{\text{joint}}(\nu|x,\{y_i\}_{1:K}) = \frac{\pi^*(y_\nu|x)/\mu(y_\nu|x)}{\sum_{j=1}^{K}\pi^*(y_j|x)/\mu(y_j|x)}.$$

$$
\begin{aligned}
p(\mathcal{O}=y_i|\{y\}_{1:K}) &= \frac{p(\{y\}_{1:K}|\mathcal{O}=y_i)p(\mathcal{O}=y_i)}{\sum_{j=1}^{K}p(\{y\}_{1:K}|\mathcal{O}=y_j)p(\mathcal{O}=y_j)} \\
&= \frac{\pi^*(y_i|x)/\mu(y_i|x)}{\sum_{j=1}^{K}\pi^*(y_j|x)/\mu(y_j|x)} \\
&= \frac{e^{r(y_i)/\alpha}}{\sum_{j=1}^{K}e^{r(y_j)/\alpha}}
\end{aligned}
$$

Define model policy as

$$\pi_\theta(y|x) := \mu(y|x)\frac{e^{r_\theta(x,y)}}{Z_\theta(x)}.$$

The model posterior probability satisfies

$$p_\theta^{\text{joint}}(\nu|x,\{y_i\}_{1:K}) = \frac{e^{r_\theta(x,y_\nu)}}{\sum_{i=1}^{K}e^{r_\theta(x,y_i)}}.$$

**Theorem A.1** (InfoNCA Objective). *For any $K>1$, $\alpha>0$, we have the following results.*

*(a) Equivalent objective.*

$$\min_\theta \mathbb{E}_{p^{\text{joint}}(x,\{y_i\})}D_{\text{KL}}[p^{\text{joint}}(\nu|x,\{y_i\})||p_\theta^{\text{joint}}(\nu|x,\{y_i\})]$$

$$\iff \min_\theta -\mathbb{E}_{p(x)\prod\mu(y_i|x)}\sum_{i=1}^{K}\frac{e^{r(x,y_i)/\alpha}}{Z(x)}\log\frac{e^{r_\theta(x,y_i)}}{\sum_{j=1}^{K}e^{r_\theta(x,y_j)}}, \tag{Eq. 5}$$

*where $Z(x) = \mathbb{E}_{\mu(y|x)}e^{r(x,y)/\alpha}$.*

*(b) Optimal solution. Assume unlimited model capacity and data samples. The optimal $r_{\theta^*}$ and $\pi_{\theta^*}$ for solving Eq. 5 are*

$$r_{\theta^*}(x,y) = r(x,y)/\alpha + C(x),$$

$$\text{and}\quad \pi_{\theta^*}(x,y) \propto \mu(y|x)e^{r(x,y)/\alpha},$$

*where $C(x)$ is an arbitrary function conditioning on $x$.*

*Proof.* **(a) Equivalent objective.**

$$\min_\theta \mathbb{E}_{p^{\text{joint}}(x,\{y_i\})} D_{\text{KL}}[p^{\text{joint}}(\nu|x,\{y_i\})||p_\theta^{\text{joint}}(\nu|x,\{y_i\})]$$

$$\Leftrightarrow \min_\theta \mathbb{E}_{p^{\text{joint}}(x,\{y_i\})} \mathbb{E}_{p^{\text{joint}}(\nu|x,\{y_i\})} \log \frac{p^{\text{joint}}(\nu|x,\{y_i\})}{p_\theta^{\text{joint}}(\nu|x,\{y_i\})}$$

$$\Leftrightarrow \min_\theta -\mathbb{E}_{p^{\text{joint}}(x,\{y_i\})} \mathbb{E}_{p^{\text{joint}}(\nu|x,\{y_i\})} \log p_\theta^{\text{joint}}(\nu|x,\{y_i\})$$

$$\Leftrightarrow \min_\theta -\mathbb{E}_{p(x)p(\nu)p^{\text{joint}}(\{y_i\}|x,\nu)} \log p_\theta^{\text{joint}}(\nu|x,\{y_i\}) \qquad \text{(Bayes' rule)}$$

$$\Leftrightarrow \min_\theta -\mathbb{E}_{p(x)p(\nu)\prod_{i=1}^K \mu(y_i|x)} \frac{\pi^*(y_\nu|x)}{\mu(y_\nu|x)} \log p_\theta^{\text{joint}}(\nu|x,\{y_i\}) \qquad \text{(importance sampling)}$$

$$\Leftrightarrow \min_\theta -\mathbb{E}_{p(x)\prod_{i=1}^K \mu(y_i|x)} \left[\mathbb{E}_{p(\nu)} \frac{\pi^*(y_\nu|x)}{\mu(y_\nu|x)} \log p_\theta^{\text{joint}}(\nu|x,\{y_i\})\right]$$

$$\Leftrightarrow \min_\theta -\mathbb{E}_{p(x)\prod_{i=1}^K \mu(y_i|x)} \left[\frac{1}{K}\sum_{\nu=1}^K \frac{\pi^*(y_\nu|x)}{\mu(y_\nu|x)} \log p_\theta^{\text{joint}}(\nu|x,\{y_i\})\right]$$

$$\Leftrightarrow \min_\theta -\mathbb{E}_{p(x)\prod_{i=1}^K \mu(y_i|x)} \left[\sum_{\nu=1}^K \frac{e^{r(x,y_\nu)/\alpha}}{Z(x)} \log p_\theta^{\text{joint}}(\nu|x,\{y_i\})\right] \qquad \text{(based on Eq. 2)}$$

$$\Leftrightarrow \min_\theta -\mathbb{E}_{p(x)\prod_{i=1}^K \mu(y_i|x)} \left[\sum_{i=1}^K \frac{e^{r(x,y_i)/\alpha}}{Z(x)} \log \frac{e^{r_\theta(x,y_i)}}{\sum_{j=1}^K e^{r_\theta(x,y_j)}}\right] \qquad \text{(change sum index)}$$

**(b) Optimal solution.**

Given conclusions from (a). With unlimited model capacity, $p_\theta^{\text{joint}}(\nu|x,\{y_i\})$ could represent any discrete distribution, such that we can arrive at the global optimal point given infinite training data.

$$\mathbb{E}_{p^{\text{joint}}(x,\{y_i\})} D_{\text{KL}}[p^{\text{joint}}(\nu|x,\{y_i\})||p_{\theta*}^{\text{joint}}(\nu|x,\{y_i\})] = 0$$

$$\implies p^{\text{joint}}(\nu|x,\{y_i\}) = p_{\theta*}^{\text{joint}}(\nu|x,\{y_i\}) \quad \forall x,\nu,\{y_i\}_{1:K}$$

$$\implies \frac{\pi^*(y_\nu|x)/\mu(y_\nu|x)}{\sum_{i=1}^K \pi^*(y_i|x)/\mu(y_i|x)} = \frac{e^{r_{\theta*}(x,y_\nu)}}{\sum_{i=1}^K e^{r_{\theta*}(x,y_i)}} \quad \forall x,\nu,\{y_i\}_{1:K}$$

$$\implies r_{\theta*}(x,y) = r(x,y)/\alpha + C(x) \quad \forall x,y,C$$

$$\implies \pi_{\theta*}(x,y) \propto \mu(y|x)e^{r(x,y)/\alpha} \quad \forall x,y$$

$\square$

## A.2 NCA Objective

Recall the optimal language policy is

$$\pi^*(y|x) = \mu(y|x)\frac{e^{r(x,y)/\alpha}}{Z(x)}. \qquad \text{(Eq. 2)}$$

Consider a response $y$ randomly sampled from either the optimal LM $\pi^*(y|x) = \mu(y|x)\frac{e^{r(x,y)/\alpha}}{Z(x)}$, or the pretrained LM $\mu(y|x)$. Let a binary variable $\nu = 1$ indicates the response $y$ is sampled from $\pi^*$. Then the marginal distribution of $y$ is

$$p^{\text{joint}}(y|x) := p(\nu = 0)\mu(y|x) + p(\nu = 1)\pi^*(y|x).$$

Given the prior $p(\nu = 0) = p(\nu = 1) = \frac{1}{2}$, using Bayes' Rule, the data posterior satisfies

$$p^{\text{joint}}(\nu = 0|x,y) = \frac{\mu(y|x)}{\mu(y|x) + \pi^*(y|x)}.$$

$$p^{\text{joint}}(\nu = 1|x, y) = \frac{\pi^*(y|x)}{\mu(y|x) + \pi^*(y|x)}.$$

Define model policy as $\pi_\theta(y|x) := \mu(y|x)e^{r_\theta(x,y)}$. The model posterior probability satisfies

$$p_\theta^{\text{joint}}(\nu = 1|x, y) = \sigma(r_\theta(x, y)).$$

$$p_\theta^{\text{joint}}(\nu = 0|x, y) = 1 - \sigma(r_\theta(x, y)) = \sigma(-r_\theta(x, y)). \tag{Eq. 8}$$

**Theorem A.2** (NCA Objective). *For any $\alpha > 0$, we have the following results.*

*(a) Equivalent objective.*

$$\min_\theta \mathbb{E}_{p^{joint}(x,y)} D_{\text{KL}}[p^{joint}(\nu|x,y)||p_\theta^{joint}(\nu|x,y)]$$

$$\iff \min_\theta -\mathbb{E}_{p(x)\mu(y|x)} \frac{e^{r(x,y)/\alpha}}{Z(x)} \log\sigma(r_\theta(x,y)) + \log\sigma(-r_\theta(x,y)), \tag{Eq. 9}$$

*where $Z(x) = \mathbb{E}_{\mu(y|x)}e^{r(x,y)/\alpha}$.*

*(b) Optimal solution. Assume unlimited model capacity and data samples. The optimal $r_{\theta^*}$ and $\pi_{\theta^*}$ for solving Eq. 5 are*

$$r_{\theta^*}(x, y) = r(x, y)/\alpha - \log\mathbb{E}_{\mu(y|x)}e^{r(x,y)/\alpha},$$

$$and \quad \pi_{\theta^*}(x, y) \propto \mu(y|x)e^{r(x,y)/\alpha}.$$

*Proof.* **(a) Equivalent objective.**

$$\min_\theta \mathbb{E}_{p^{\text{joint}}(x,y)} D_{\text{KL}}[p^{\text{joint}}(\nu|x,y)||p_\theta^{\text{joint}}(\nu|x,y)]$$

$$\Leftrightarrow \min_\theta \mathbb{E}_{p^{\text{joint}}(x,y)}\mathbb{E}_{p^{\text{joint}}(\nu|x,y)} \log\frac{p^{\text{joint}}(\nu|x,y)}{p_\theta^{\text{joint}}(\nu|x,y)}$$

$$\Leftrightarrow \min_\theta -\mathbb{E}_{p^{\text{joint}}(x,y)}\mathbb{E}_{p^{\text{joint}}(\nu|x,y)} \log p_\theta^{\text{joint}}(\nu|x,y)$$

$$\Leftrightarrow \min_\theta -\mathbb{E}_{p(x)p(\nu)p^{\text{joint}}(y|x,\nu)} \log p_\theta^{\text{joint}}(\nu|x,y) \qquad \text{(Bayes' rule)}$$

$$\Leftrightarrow \min_\theta -[p(\nu = 0)\mathbb{E}_{p(x)p^{\text{joint}}(y|x,\nu=0)} \log p_\theta^{\text{joint}}(\nu = 0|x, y)+$$

$$p(\nu = 1)\mathbb{E}_{p(x)p^{\text{joint}}(y|x,\nu=1)} \log p_\theta^{\text{joint}}(\nu = 1|x, y)]$$

$$\Leftrightarrow \min_\theta -\left[\mathbb{E}_{p(x)\mu(y|x)} \log\sigma(-r_\theta(x,y)) + \mathbb{E}_{p(x)\pi^*(y|x)} \log\sigma(r_\theta(x,y))\right] \qquad \text{(by Eq. 8)}$$

$$\Leftrightarrow \min_\theta -\mathbb{E}_{p(x)\mu(y|x)} \frac{e^{r(x,y)/\alpha}}{Z(x)} \log\sigma(r_\theta(x,y)) + \log\sigma(-r_\theta(x,y)) \qquad \text{(importance sampling)}$$

**(b) Optimal solution.**

Given conclusions from (a). With unlimited model capacity, $p_\theta^{\text{joint}}(\nu|x, y)$ could represent any discrete distribution, such that we can arrive at the global optimal point given infinite training data.

$$\mathbb{E}_{p^{\text{joint}}(x,y)} D_{\text{KL}}[p^{\text{joint}}(\nu|x,y)||p_{\theta^*}^{\text{joint}}(\nu|x,y)] = 0$$

$$\implies p^{\text{joint}}(\nu|x, y) = p_{\theta^*}^{\text{joint}}(\nu|x, y) \quad \forall x, \nu, y$$

$$\implies p^{\text{joint}}(\nu = 1|x, y) = p_{\theta^*}^{\text{joint}}(\nu = 1|x, y) \quad \forall x, y$$

$$\implies \frac{\pi^*(y|x)}{\mu(y|x) + \pi^*(y|x)} = \sigma(r_\theta^*(x, y)) = \frac{e^{r_\theta^*(x,y)}}{1 + e^{r_\theta^*(x,y)}} \quad \forall x, y$$

$$\implies e^{r_\theta^*(x,y)} = \frac{\pi^*(y|x)}{\mu(y|x)} \quad \forall x, y$$

$$\implies r_{\theta^*}(x, y) = r(x, y)/\alpha - \log\mathbb{E}_{\mu(y|x)}e^{r(x,y)/\alpha} \quad \forall x, y$$

$$\implies \pi_{\theta^*}(x, y) \propto \mu(y|x)e^{r(x,y)/\alpha} \quad \forall x, y$$

$\square$

## B  Pseudocode

PyTorch code for the InfoNCA/NCA loss for reward datasets is provided below:

```python
import torch.nn.functional as F

def reward_loss(pi_logps, ref_logps, rewards, alpha, beta, loss_type):
    """
    pi_logps: policy logprobs for K responses, shape (B, K)
    ref_logps: reference logprobs for K responses, shape (B, K)
    rewards: reward labels for K responses, shape (B, K)
    alpha: the reward temperature controlling strength of KL penalty
    beta: the parameterization coefficient that defines the reward model
    loss_type: could be either "InfoNCA" or "NCA" loss
    """

    soft_labels = (rewards / alpha).softmax(dim=-1) # (B, K)

    model_rewards = (pi_logps - ref_logps) * beta # (B, K)

    if loss_type == "InfoNCA":
        model_logps = model_rewards.log_softmax(dim=-1) # (B, K)
        losses = - (soft_labels * model_logps).sum(dim=-1) # (B,)
    elif loss_type == "NCA":
        optimization = - (soft_labels * F.logsigmoid(model_rewards)).sum(dim=-1) # (B,)
        regularization = - F.logsigmoid(-model_rewards).mean(dim=-1) # (B,)
        losses =  optimization + regularization # (B,)

    return losses.mean()
```

The loss implementation under pairwise preference settings is equivalent to reward losses with $K = 2$ and $\alpha \to 0$. We provide the code separately for easy comparison with DPO.

```python
def preference_loss(chosen_pi_logps, chosen_ref_logps,
                    rejected_pi_logps, rejected_ref_logps,
                    beta, loss_type):
    """
    chosen_pi_logps: policy logprobs for the preferred responses, shape (B, )
    chosen_ref_logps: reference logprobs for the preferred responses, shape (B, )
    rejected_pi_logps: policy logprobs for the dispreferred responses, shape (B, )
    rejected_ref_logps: reference logprobs for the dispreferred responses, shape (B, )
    beta: the parameterization coefficient that defines the reward model
    loss_type: one of "InfoNCA", "NCA" or "DPO" loss
    """

    chosen_rewards = (chosen_pi_logps - chosen_ref_logps) * beta # (B,)
    rejected_rewards = (rejected_pi_logps - rejected_ref_logps) * beta # (B,)

    if loss_type in ["DPO", "InfoNCA"]:
        losses = -F.logsigmoid(chosen_rewards - rejected_rewards) # (B,)
    elif loss_type == "NCA":
        losses = - F.logsigmoid(chosen_rewards) \
                 - 0.5 * F.logsigmoid(-chosen_rewards) \
                 - 0.5 * F.logsigmoid(-rejected_rewards) # (B,)

    return losses.mean()
```

# C   Experimental Details

**Experiments with UltraFeedback.**  Our implementation is heavily based on the Transformer Reinforcement Learning (TRL) library [41] and Zephyr's official code base [40]. All models are fine-tuned from the publicly accessible HuggingFaceH4/mistral-7B-SFT-beta model. Experiments are run on Nvidia A40 or RTX 4090 GPUs using bfloat16 precision. We ablate $\beta \in \{3e-4, 1e-3, 3e-3, 1e-2, 3e-2, 1e-1, 3e-1, 1.0\}$ and $\alpha \in \{0.01, 0.1, 0.33, 1.0, 3.33\}$. The default reward temperature $\alpha$ is 0.01. The default parameterization coefficient $\beta$ is also 0.01. We adopt the QLoRA [10] fine-tuning technique with rank 16, $\alpha_{\text{lora}} = 16$, and a dropout rate of 0.05. We train all models for 1 epoch. The batch size is 32. We use an AdamW optimizer with a learning rating of 5e-6. For KTO and IPO baselines, we adopt exactly the same training pipeline for reporting their performance except that we tune the $\beta \in \{0.01, 0.1, 0.3, 0.5, 1.0\}$. We find the most suitable $beta$ for KTO is 0.01, and for IPO is 0.5.

**Experiments with UltraInteract.**  We follow [44] and fine-tune all parameters of the pretrained model with UltraInteract and UltraFeedback. Specifically, for HuggingFaceH4/mistral-7B-SFT-beta model, we directly adopt the recipe of Eurus, with the same data mixture (all 220K multi-turn trajectory pairs in UltraInteract and all 340K pairs in UltraFeedback) and hyperparameters. All $\beta$ is set to 0.1. The model is trained for 1 epoch with a cosine schedule. The learning rate is 5e-7 and the warmup ratio is 0.1. For Mixtral-8x7B, we first SFT the model with the recipe for Eurus-70B-SFT, and then applied the same preference learning recipe as aforementioned.

Figure 7: Rating system screenshot. Our rating system is adapted from [47].

# D  Additional Experiment Results

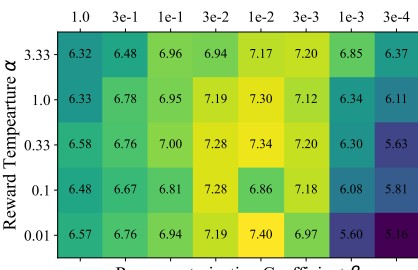 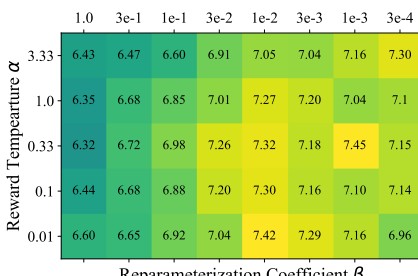

Figure 8: MT-bench score for **InfoNCA (left)** and **NCA (right)** regarding various $\alpha$ and $\beta$. Results are averaged for $K = 2$ and $K = 4$ experiments. Overall, model performance is relatively more sensitive to variations of $\beta$ compared with $\alpha$. NCA shows greater tolerance to hyperparameter variations.

| Method | Functional Form | MT-bench | KL Divergence |
|---|---|---|---|
| Mistral-7B-SFT | $-$ | 6.45 | 0.00 |
| +DPO | $r_\theta = \beta \log \frac{\pi_\theta}{\mu}$ | 7.34 | 0.53 |
| +InfoNCA | $r_\theta = \beta \log \frac{\pi_\theta}{\mu}$ | **7.63** | **0.47** |
|  | $r_\theta = \beta \log \pi_\theta$ | 4.65 | 4.39 |
|  | $r_\theta = \beta \log \pi_\theta$ + SFT regularization | 7.15 | 0.26 |
| +NCA | $r_\theta = \beta \log \frac{\pi_\theta}{\mu}$ | **7.52** | **0.39** |
|  | $r_\theta = \beta \log \pi_\theta$ | 5.82 | 3.32 |
|  | $r_\theta = \beta \log \pi_\theta$ + SFT regularization | 6.88 | 0.09 |

Table 4: Comparison of different functional forms of $r_\theta$ on UltraFeeback datasets.

# E   Appendix F: Additional Empirical Results

We provide sampled responses of LM policies trained respectively with the DPO and NCA to compare their data regularization feature. In general, generations from the NCA policy are more similar to the preferred response in the dataset.

| **Prompt** |
| --- |
| You are given an answer. Create a question for the answer.
Grams to Ounces conversion 1 gram (g) is equal to 0.03527396195 ounces (oz). |
| **Preferred Response in Dataset** (r=8.0) |
| What is the conversion rate for grams to ounces for 1 gram? |
| **Dispreferred Response in Dataset** (r=7.0) |
| What is the relationship between grams and ounces in weight measurement? |
| **Reponse generated by** $\pi_\theta^{\mathbf{DPO}}$ (r=2.0) |
| How many ounces are equivalent to 200 grams according to the given conversion rate of 1 gram (g) to 0.03527396195 ounces (oz)? |
| **Reponse generated by** $\pi_\theta^{\mathbf{NCA}}$ (r=10.0) |
| Question: What is the conversion rate from grams to ounces, and how many grams are one ounce? |

Table 5: Sampled generations and the log-likelihood ratio of the NCA and the DPO/InfoNCA policy.

| **Prompt** |
| --- |
| If "A busy crosswalk somewhere in asia." does that mean that "The crosswalk was blocked off by police."?
Options:
- yes; - it is not possible to tell ; - no |
| **Preferred Response in Dataset** (r=8.0) |
| It is not possible to tell.
Confidence: 95% |
| **Dispreferred Response in Dataset** (r=7.5) |
| Question:
If "A busy crosswalk somewhere in asia." does that mean that "The crosswalk was blocked off by police."?
Answer:
It is not possible to tell.
Confidence: 90% |
| **Reponse generated by** $\pi_\theta^{\mathbf{DPO}}$ (r=3.0) |
| - It is not possible to determine whether "The crosswalk was blocked off by police." in "A busy crosswalk somewhere in Asia." To determine whether the crosswalk was blocked off by police, you would need additional information or context. |
| **Reponse generated by** $\pi_\theta^{\mathbf{NCA}}$ (r=8.0) |
| It is not possible to tell.
Confidence: 95% |

Table 6: Sampled generations and the log-likelihood ratio of the NCA and the DPO/InfoNCA policy.

**Prompt**

In this task, you need to count the occurrences of the given word in the given sentence.
Sentence: 'a green double decker bus sitting on top of a parking lot'. Count the occurrences of the word 'a' in the given sentence.

**Preferred Response in Dataset** (r=9.0)

The word 'a' appears 3 times in the given sentence.
Confidence: 100%

**Dispreferred Response in Dataset** (r=6.0)

The given sentence is "a green double decker bus sitting on top of a parking lot." To answer your question about the occurrences of the word 'a', I will count the number of times the word appears in the sentence.
After counting the occurrences of the word 'a' in the sentence, I found that it appears once. Therefore, the answer to your question is that the word 'a' appears once in the given sentence.

**Reponse generated by $\pi_\theta^{\mathbf{DPO}}$** (r=4.0)

Of course, I'm here to help! There are 11 occurrences of the word 'a' in the given sentence. Here's how you can count them: 1. 'a' in 'a' 2. 'a' in 'green' 3. 'a' in 'double' 4. 'a' in 'decker' 5. 'a' in 'bus' 6. 'a' in 'sitting' 7. 'a' in 'on ...
Confidence: 90%

**Reponse generated by $\pi_\theta^{\mathbf{NCA}}$** (r=8.0)

Sure! The word 'a' appears 3 times in the given sentence.

Table 7: Sampled generations and the log-likelihood ratio of the NCA and the DPO/InfoNCA policy.

**Prompt**

if $X^2 + 3 = 12$. What is X?

**Preferred Response in Dataset** (r=9.0)

To find the value of X, we'll first need to isolate $X^2$:
$X^2 + 3 = 12$
Subtract 3 from both sides:
$X^2 = 9$
Now, we can find the square root of both sides of the equation to solve for X:
$X = \pm 3$
So, X can be either 3 or -3.

**Dispreferred Response in Dataset** (r=4.0)

I'm sorry, I cannot solve this algebra equation. Can I assist you with anything else?

**Reponse generated by $\pi_\theta^{\mathbf{DPO}}$** (r=7.0)

Sure, I'd love to help. By solving the equation, the value of X should be 3.

**Reponse generated by $\pi_\theta^{\mathbf{NCA}}$** (r=9.0)

To find the value of $X$, we need to solve the equation $X^2 + 3 = 12$.
First, subtract 3 from both sides:
$X^2 = 12 - 3$ $X^2 = 9$
Next, take the square root of both sides:
$X = \pm\sqrt{9}$
So, $X$ can be either 3 or $-3$.

Table 8: Sampled generations and the log-likelihood ratio of the NCA and the DPO/InfoNCA policy.

## Appendix H: Comparing NCA with DPO on Data Regularization.

Suppose there are only 4 possible responses $\{y_a, y_b, y_c, y_d\}$. Each response has a reward $r$, and a pretrained language model policy $\mu$:

| - | $y_a$ | $y_b$ | $y_c$ | $y_d$ |
|---|---|---|---|---|
| Likelihood $\mu(y)$ | 40% | 50% | 5% | 5% |
| Reward $r(y)$ | 10 | 3 | 7 | 0 |

Table 9: Initial probabilities and rewards

Given a preference dataset $D = \{y_a > y_b\}$ ($y_c$ and $y_d$ do not exist in the dataset), the loss functions are defined as:

$$L_\theta^{\text{DPO}} = -\log \sigma \left( \log \frac{\pi_\theta(y_a)}{\mu(y_a)} - \log \frac{\pi_\theta(y_b)}{\mu(y_b)} \right)$$

$$L_\theta^{\text{NCA}} = -\log \sigma \left( \log \frac{\pi_\theta(y_a)}{\mu(y_a)} \right) - \frac{1}{2} \log \sigma \left( -\log \frac{\pi_\theta(y_a)}{\mu(y_a)} \right) - \frac{1}{2} \log \sigma \left( -\log \frac{\pi_\theta(y_b)}{\mu(y_b)} \right)$$

After fine-tuning, there are several possibilities for $\pi_\theta$:

| - | $\pi_\theta(y_a)$ | $\pi_\theta(y_b)$ | $\pi_\theta(y_c)$ | $\pi_\theta(y_d)$ | $\log \frac{\pi_\theta(y_a)}{\mu(y_a)} - \log \frac{\pi_\theta(y_b)}{\mu(y_b)}$ | $\bar{r}$ |
|---|---|---|---|---|---|---|
| (1) | 20% $\downarrow$ | 10% $\downarrow\downarrow$ | 5% | 65% $\uparrow$ | 0.916 > 0 | 2.65 $\downarrow$ |
| (2) | 20% $\downarrow$ | 10% $\downarrow\downarrow$ | 65% $\uparrow$ | 5% | 0.916 > 0 | 6.85 $\uparrow$ |
| (3) **Wanted** | 60% $\uparrow$ | 30% $\downarrow$ | 5% | 5% | 0.916 > 0 | 7.25 $\uparrow$ |

Table 10: Post fine-tuning probabilities and rewards

In scenarios (1) and (2), we can see that the likelihood for both $y_a$ and $y_b$ decreases. However, (1) and (2) satisfy the DPO loss function because the likelihood for $y_b$ decreases more, and the relative likelihood margin between $y_a$ and $y_b$ becomes larger. In (1) and (2), the likelihood for either $y_c$ or $y_d$ increases because $\pi(y_a) \downarrow + \pi(y_b) \downarrow + \pi(y_c)? + \pi(y_d)? = 1$. However, $y_c$ and $y_d$ are unreliable because we do not know their quality (rewards). The LM policy could generalize to a low-quality response like $y_d$ (case (1)).

In contrast, the NCA effectively prevents the winning response likelihood $\pi(y_a)$ from decreasing, because it mainly optimizes the absolute data likelihood instead of just caring about the relative likelihood margin $\log \frac{\pi_\theta(y_a)}{\mu(y_a)} - \log \frac{\pi_\theta(y_b)}{\mu(y_b)}$. Thus, we say NCA is more likely to assign a larger likelihood to responses within the dataset.

## Appendix G: Comparision with Related Works

We compare with a prior work SLiC-HF [45] in this section. SLiC is inspired by [46] and similarly aims to calibrate sequence likelihood to align with human preferences. Given a preference data pair $\{x \to y_w > y_l\}$, the loss function for SLiC is

$$L_\theta := \max(0, \delta - \log \pi_\theta(y_w|x) + \log \pi_\theta(y_w|x)) - \lambda \log \pi_\theta(y_w|x),$$

where $\delta$ is a hyperparameter that controls the likelihood margin of data, and $\lambda$ controls the regularization weight of the loss. The main difference between our proposed method and the SLiC loss can be summarized as follows:

- **Theoritical framework.** SLiC is mainly adapted from the existing LM calibration methods [46]. In contrast, our method is based on noise contrastive estimation methods [14, 24].

- **Policy regularization.** The training process of SLiC is regularized by the additional SFT loss controlled by $\lambda$. In contrast, our proposed method is regularized through the parameterization technique $r_\theta = \beta \log \frac{\pi_\theta}{\mu}$ controlled by $\beta$.

- **Learning target.** SLiC directly optimizes the policy model $\pi_\theta$, while our method directly optimizes the residual model $r_\theta$.

# F   Limitations and Broder Impacts

**Limitations.**   We mainly discuss the theoretical limitations of our methods. Although both InfoNCA and NCA enjoy strong theoretical guarantees that ensure convergence to the optimal LM distribution, these guarantees only hold given infinite response numbers per instruction ($K \rightarrow \infty$) during training to ensure an accurate estimate of $Z(x)$, which is clearly unrealistic. Additionally, while more responses employed during training improve performance, they also require higher GPU memory consumption, leading to a trade-off between performance and efficiency.

**Broder Impacts.**   This research contributes to the field of machine learning by enhancing language model alignment with human preferences. While these improvements can lead to more accurate and ethically aligned AI-generated content, they also pose ethical challenges. Notably, there is a risk of reinforcing biases present in data or human preferences. Additionally, the increasing capabilities of language models may impact employment in sectors reliant on language skills. It is crucial to continually assess these ethical considerations and societal impacts, ensuring that advancements in machine learning are guided by responsible and inclusive practices.

