# OpenReview forum: "Noise Contrastive Alignment of Language Models with Explicit Rewards"
_NeurIPS.cc/2024/Conference — NeurIPS 2024 poster_

### Official Review · Reviewer_xAZC · 2024-07-13

**Soundness:** 3
**Presentation:** 3
**Contribution:** 3
**Rating:** 7
**Confidence:** 4

**Summary:**

The paper's main contributions are:

1. Theoretical Integration: It integrates Direct Policy Optimization (DPO) with contrastive learning theories, offering a general framework for using reward and preference data.
2. Value of Suboptimal Responses: Highlights the importance of suboptimal responses in optimizing language models, showing improved performance over other methods by fully utilizing reward data.
3. Performance Improvement: Demonstrates how NCA counters the data likelihood decline in DPO, enhancing practical performance.

**Strengths:**

1. Utilizing the explicit reward for alignment which is quite interesting.

2. Propose a general method to address this problem.

3. It sees improvement over DPO on leaderboards. Specially, it is much better than DPO in math and coding tasks.

**Weaknesses:**

I don't have further suggestions for this part.

**Questions:**

How do you compare your work with the paper titled Direct Preference Optimization with an Offset(https://arxiv.org/pdf/2402.10571).

**Limitations:**

I don't see any issues with this part.

---

> ### Author Rebuttal · Authors · 2024-08-01
>
> # Official Response to Reviewer xAZC
> We thank reviewer xAZC for the valuable feedback. It really encourages us to see that the reviewer finds our work to be a good contribution to the field, regarding pointing out the value of suboptimal responses and countering the likelihood decline trend in alignment. We answer the reviewer's question below:
>
> **Q1: How do you compare your work with the paper titled Direct Preference Optimization with an Offset ([https://arxiv.org/pdf/2402.10571](https://arxiv.org/pdf/2402.10571)).**
>
>
> **A1:** We thank the reviewer for mentioning this paper. ODPO also wants to leverage the reward gap information for better optimization. We notice that ODPO is followed by some recent work like SimPO due to its effectiveness, which reviewer UYN1 has mentioned.
>
> In simple words, ODPO modifies the DPO loss function to be
> $L_\theta^\text{ODPO}  =  - \log \sigma(r_\theta(x,y_w) - r_\theta(x,y_l)  - \gamma)$,
> where $\gamma$ is determined by the reward gap in datasets.
>
> Connection with our methods:
> 1. InfoNCA and ODPO share similar motivations but have distinctive approaches. In our understanding, ODPO can only guarantee the learned reward gap **is larger** than $\gamma$, while InfoNCA converges if and only if the learned reward gap is **equal to** $\gamma$.
> 2. InfoNCA can be directly used for multiple responses while ODPO is designed for pairwise responses.
> 3. Based on our experience, the above two distinctions won't cause very much empirical difference in practice regarding language quality. We personally feel InfoNCA loss is more mathematically elegant, though. After all, it is our own method :)
> 4. Compared with NCA, ODPO is optimizing relative rewards just like DPO, so we guess the problem of logp decline still exists. NCA may perform better in preventing the logp decline trend.

---

### Official Review · Reviewer_WFsu · 2024-07-15

**Soundness:** 3
**Presentation:** 3
**Contribution:** 3
**Rating:** 7
**Confidence:** 3

**Summary:**

## Summary

- Typically, RLHF algorithms like DPO use preference pairs dataset.
- The key question, this paper tries to answer is how to incorporate reward datasets annotated with scalar values. Previous approach usually prune the scalar reward datasets by selecting the best response and pairing with a random remaining response. However, this work shows that the extra suboptimal responses can also be beneficial for policy training.
- They present two main algorithms (inspired from contrastive learning literature) here which handle the aforementioned problem
	- NCA Noice Contrastive Alignment
	- InfoNCA
- They also show that NCA is able to mitigate the "decreasing likelihood trend" observed with the DPO algorithm. NCA also outperforms DPO on math and coding tasks because it prevents the likelihood of preferred responses from decreasing. The main difference between InfoNCA and NCA objective is that the NCA objective has an additional regularizer term
	- The "decreasing likelihood trend" can be simply described as the decrease in likelihood of the preferred response after DPO training.
	- This happens because DPO focuses on adjusting relative likelihoods across generations whereas NCA focuses on adjusting absolute likelhood
- InfoNCA compares multiple responses and identifies the one sampled from optimal policy. NCA predicts the model source of a single response.
- The authors also show that DPO is a special case of InfoNCA

**Strengths:**

## Strengths
- The loss objectives are intuitively easy to follow from equation 6 and equation 11.
- The paper is well written and easy to follow, with clear and accessible writing. The literature review provides a solid contextual foundation for the work.
- The experiments do a good job at substantiating the claims presented and justifying the NCA and InfoNCA algorithm.
- I also like the section on empirical takeaway: when to choose NCA over DPO.
- The paper overall addresses an important problem and proposes the NCA solution which helps tackle the decreasing likelihood issue. The empirical takeaway for reasoning tasks is also helpful for an ML practitioner. My overall recommendation for the paper is an accept but I would very much appreciate if the authors can answer my questions to help deepen my understanding of the work.

**Weaknesses:**

I could not find any weakness

**Questions:**

## Questions

- I have not gone through the material in the appendix but at several places in the theorems, the authors assume a model with unlimited capacity. Why is this important and what are the implications otherwise?
- It is not obvious to me how equation 11 leads to predicting the model source of a single response in Figure 3

**Limitations:**

The authors have adequately addressed the limitations.

---

> ### Author Rebuttal · Authors · 2024-08-06
>
> # Official Response to Reviewer WFsu
>
> We appreciate the reviewer WFsu for the very detailed comments on our paper and are glad that the reviewer finds our paper to be helpful. Below we address the reviewer's questions and hope this can help the reviewer increase the confidence of our work.
>
> **Q1: The paper assumes a model with unlimited capacity. Why is this important and what are the implications otherwise?**
>
> **A1:** We assume unlimited model capacity mainly because we want our proof to be strict. There's nothing special here.
>
> Imagine we want to fix a function $f(x)=x^2$ using neural networks. We must first assume our model is expressive enough (unlimited capacity) to fully achieve convergence. Otherwise, if our model $f_\theta$ is simply a linear MLP (limited capacity), we cannot achieve $f_\theta(x)=x^2$ no matter how hard we try.
>
> **Q2:  It is not obvious to me how equation 11 leads to predicting the model source of a single response in Figure 3**
>
> **A2:** Eq 11 is the NCA loss function. For convenience, we suppose there are only $K=2$ responses $y_1$ and $y_2$, with their respective reward $r_1 > r_2$.
>
> $L_\theta^\text{NCA}  =- \sum_{i=1}^2 \left[\frac{e^{r_i}}{e^{r_1}+e^{r_2}} \log \sigma (r_\theta(x,y_i)) + \frac{1}{2} \log \sigma (-r_\theta(x,y_i))\right]$
>
> We compare with $L^\text{InfoNCA}_\theta$ for reference:
>
> $L_\theta^\text{InfoNCA}  = - \sum_{i=1}^2 \left[\frac{e^{r_i}}{e^{r_1}+e^{r_2}} \log \frac{e^{r_\theta(x,y_i)}}{e^{r_\theta(x,y_1)}+e^{r_\theta(x,y_2)}}\right]$
>
>
> According to our proofs in the paper, one optimal solution for the above training loss is that
>
> $r_\theta(x,y_1) = r_1$ and $r_\theta(x,y_2) = r_2$.
>
> However, if we let $r_\theta(x,y_1) = r_1 -12345.0$ and  $r_\theta(x,y_2) = r_2 -12345.0$, we will find this is still the optimal solution for InfoNCA.
>
> $- \sum_{i=1}^2 \left[\frac{e^{r_i}}{e^{r_1}+e^{r_2}} \log \frac{e^{r_\theta(x,y_i) - 12345.0}}{e^{r_\theta(x,y_1) - 12345.0}+e^{r_\theta(x,y_2) - 12345.0}}\right]=- \sum_{i=1}^2 \left[\frac{e^{r_i}}{e^{r_1}+e^{r_2}} \log \frac{e^{r_\theta(x,y_i)}}{e^{r_\theta(x,y_1)}+e^{r_\theta(x,y_2)}}\right]$
>
> **This is exactly why we say InfoNCA only targets relative rewards.** The loss function is happy as long as $r_\theta(x,y_1) - r_\theta(x,y_2)$>0. However, $r_\theta(x,y_1)$ can be a very negative number, which we do not want.
>
> **For the NCA loss, it is otherwise**, because inside $\sigma$ there is only a single reward  $\sigma(r_\theta(x,y_i))$ instead of many **relative** model rewards $\log \frac{e^{r_\theta(x,y_i)}}{e^{r_\theta(x,y_1)}+e^{r_\theta(x,y_2)}}$.
>
> $L_\theta^\text{NCA}  =  L_\theta^\text{NCA}(x,y_1) +L^\text{NCA}_\theta(x,y_2)$
>
> $L_\theta^\text{NCA}(x,y_1) =- \left[\frac{e^{r_1}}{e^{r_1}+e^{r_2}} \log \sigma (r_\theta(x,y_1)) + \frac{1}{2} \log \sigma (-r_\theta(x,y_1))\right]$
>
> $L_\theta^\text{NCA}(x,y_2) =- \left[\frac{e^{r_2}}{e^{r_1}+e^{r_2}} \log \sigma (r_\theta(x,y_2)) + \frac{1}{2} \log \sigma (-r_\theta(x,y_2))\right]$
>
> $(r_1>r_2,  \frac{e^{r_1}}{e^{r_1}+e^{r_2}} > \frac{1}{2})$
>
> You can see that $L_\theta^\text{NCA}(x,y_1)$ is only influenced by $r_\theta(y_1)$ and is unaffected by $r_\theta(y_2)$.  For NCA, if $r_\theta(x,y_1) = r_1 -12345.0$ is a very negative number, $L_\theta^\text{NCA}(x,y_1)$ would surely be very large and contradicts the training loss minimization. **In other words, NCA forces $r_\theta(x,y_1)$ to be positive and $r_\theta(x,y_2)$ to be negative respectively and independently.** This is why we say Eq 11 leads to predicting the model source (optimize absolute reward) of a single response.
>
> We'd also like to refer the reviewer to Appendix H, where there is a more detailed example.

---

### Official Review · Reviewer_UYN1 · 2024-07-17

**Soundness:** 3
**Presentation:** 3
**Contribution:** 3
**Rating:** 7
**Confidence:** 3

**Summary:**

The authors introduce a general framework for LM alignment, leveraging Noise Contrastive Estimation (NCE) to bridge the gap in handling reward datasets explicitly annotated with scalar evaluations. The framework comprises two parallel algorithms, NCA
 and InfoNCA, both enabling the direct extraction of an LM policy from reward data as well as preference data.

**Strengths:**

1. The method bridge the theoretical gap between DPO and classic contrastive learning theories. InfoNCA and NCA are uniquely suited for both reward and preference data, offering a general framework that integrates preference-based algorithms.
2. The proposed method outperforms various preference methods by fully exploiting data information in reward datasets.
3. It's great to see the method not hurting UltraInteract and leads to better performance.

**Weaknesses:**

1. The proposed method looks trivial by integrating classic contrastive learning theories. Different methods coming out at the same time. They look similar and not easy to find which is better, such as the concurrent work SimPO: Simple Preference Optimization with a Reference-Free Reward.

**Questions:**

More training details, such as hyper-parameter search space

**Limitations:**

It would be better list more training details.

---

> ### Author Rebuttal · Authors · 2024-08-06
>
> # Official Response to Reviewer UYN1
>
> We thank reviewer UYN1 for the praise and suggestions regarding our work. We are pleased to see the reviewer's acknowledgment of our theoretical contribution in bridging the gap between DPO and classic contrastive learning theories. Below we address the reviewer's questions and hope our responses can help increase the reviewer's confidence in our work.
>
> **Q1: The proposed method looks trivial by integrating classic contrastive learning theories. Different methods coming out at the same time. They look similar and not easy to find which is better, such as the concurrent work: SimPO.**
>
> **A1:** There are indeed numerous recent approaches for solving alignment tasks, such as DPO/KTO/IPO/ORPO/SimPO, and it is challenging to definitively determine which is superior.
> Still, we would like to highlight some key features of InfoNCA/NCA that we find to be unique.
>
> 1. Unlike most previous methods, InfoNCA/NCA is not based on any kind of preference model assumptions, such as Bradley Terry or Plackett-Luce models. It is **directly derived from existing contrastive learning Bayes theories** and is deeply connected with NCE methods that have been widely validated. The theoretical cleanness should provide more confidence for the users.
> 2. Our methods, **unlike SimPO**, can handle **continuous explicit rewards** instead of just preference data. To the best of our knowledge, there are very few work/algorithms that are directly motivated by this task.
> 3. Our method guarantee **strictly convering to the target optimal policy**. The theoretical proofs are strict and purely Bayes (Appendix ). This offers a unique theoretical perspective for understanding existing alignment approaches, as reviewers EyJu and EZsK have also pointed out.
>
> **Q2: It would be better to list more training details such as hyper-parameter search space.**
>
> **A2:** We'd like to refer the reviewer to Appendix C of the paper, where we have listed all important training details (we think). Source code is also provided to ensure reproducibility.
>
> We ablate $\beta \in \{3e-4, 1e-3, 3e-3, 1e-2, 3e-2, 1e-1, 3e-1, 1.0\}$ and $\alpha \in \{0.01, 0.1, 0.33, 1.0, 3.33\}$ and $K \in {2,4}$ for ablation studies. For the main experiment Table, we all use consistent hyperparameters same as the Zephyr DPO baseline for a fair comparison and to avoid hyperparameter overfitting.
>
> We will be glad to provide any additional experiment details if the reviewer is interested in any specific aspects.

---

### Official Review · Reviewer_EZsK · 2024-07-19

**Soundness:** 2
**Presentation:** 3
**Contribution:** 3
**Rating:** 7
**Confidence:** 4

**Summary:**

This paper leverages Noise Contrastive Estimation (NCE) to align language models (LMs) with explicit reward data, which can handle both scalar evaluations and multiple preference data. The proposed methods, NCA and InfoNCA, extend current alignment theories by improving upon DPO and addressing the issue of decreasing likelihood observed in DPO. Experimental results demonstrate that InfoNCA and NCA outperform existing preference-based methods.

**Strengths:**

1. The paper bridges the gap between preference-based and reward-based alignment methods by deriving InfoNCA from InfoNCE, thereby extending DPO to handle explicit rewards with a strong theoretical foundation.
2. Empirical results show significant performance improvements in various tasks, including complex reasoning tasks, by effectively leveraging both optimal and suboptimal responses from reward datasets.
3. Mitigating Likelihood Decrease: NCA effectively prevents the chosen likelihood from decreasing during training, addressing a critical issue in DPO and enhancing model performance stability.

**Weaknesses:**

1. The motivation of this paper is weak: The description that DPO can only be used for pairwise preference data is wrong (such as Line 31 and 43). In fact, the appendix of DPO says that it can be applied to multiple preference data based on the Plackett-Luce model.
2. The main experiments lack related baselines. Since the proposed method is a reward-based method that utilizes reward data for preference ranking, Table 2 should also include PPO under the annotation type "Reward" for direct comparison since it also utilizes reward data. In addition, I suggest comparing it with PRO (https://arxiv.org/pdf/2306.17492) under the annotation type "preference" since this work directly applies DPO to the setting of multiple preference ranking.
3. The proposed method InfoNCA uses both preference data and reward data for training, resulting in more computational overhead compared with PPO and DPO. Thus, it is not surprising that better performance is achieved than with preference-based methods, although the improvement is marginal. Is it a fair comparison if the methods use different magnitudes of data? Please also see comment 1.

**Questions:**

1. What is the amount and percentage of reward and preference data in each dataset?
2. Since InfoNCA and NCA are reward-based methods, why not compare them with PPO in terms of MT-bench, AlpcaEval, and Wins vs. PPO?

**Limitations:**

Limitations are discussed in the appendix

---

> ### Author Rebuttal · Authors · 2024-08-06
>
> # Official Response to Reviewer EZsK (1/2)
>
> **Q1: The motivation is weak. The description that DPO can only be used for pairwise preference data is wrong (such as Line 31 and 43). In fact, the appendix of DPO says that it can be applied to multiple preference data based on the Plackett-Luce model.**
>
> **A1:** We thank the reviewer for the valuable feedback on the PL model. PL can indeed be combined with the DPO algorithm to handle multiple responses. Since this method is **only mentioned in the appendix** and does not have any accompanying experiments in DPO's paper, we did not notice this part originally.
>
> However, **we respectfully disagree with the reviewer regarding "weak motivation"**. The PL+DPO method **does not hurt the core contribution of our work.**  Our paper is titled "Alignment of LMs with **Explicit Rewards**". **NCA is strongly motivated** to handle explicit reward data that naturally can be applied to responses of arbitrary numbers. In contrast, like the BT model, **PL model is still a preference/ranking-based method and cannot handle reward datasets.**
>
> **We have rephrased all related descriptions** in our paper to avoid previous unstrict claims. (We cannot update our official manuscript during rebuttal due to review policy, though)
>
> For instance,
> (Line 31)
> > DPO is only tailored for preference data ~~with K = 2 responses per instruction x~~.
>
> (Line 43)
> >However, 43 unlike DPO which is built upon assumptions of ~~Bradley-Terry models~~ **preference models like Bradley-Terry or Plackett-Luce model**, InfoNCA is strictly derived from InfoNCE
>
> We have also conducted an **additional experiment** to compare with PL-based DPO methods:
>
> | Experiment| MT-bench | Alpaca | Avg. |
> |---|---|---|---|
> | DPO baseline (K=2) | 73.4 | 90.6   | 82.0 |
> | DPO × $C$ (K=4) | 73.8  | 90.3 | 82.1|
> | DPO × 3 (K=4)  | 72.2  | 91.6 | 81.9|
> | **DPO + PL Method** (K=4) | 74.1 | 91.3 | 82.7 |
> | **InfoNCA (K=4)** | **76.3** | **92.4** | **84.4** |
>
> ***
>
> **Q2: The proposed method InfoNCA uses both preference data and reward data for training, .....more computational overhead compared with PPO and DPO.....Is it a fair comparison if the methods use different magnitudes of data?  **
>
> **A2:**  We respectfully disagree with the reviewer's comment and think **there is clearly some misunderstanding of our experiments**, which we hope to clarify:
>
> 1. Our method can handle both preference and reward data, but it only requires one data type for training. As a matter of fact, we use **either** 100% of preference data **or** 100% of reward data throughout our experiments. **There is no mixed dataset used**. (Refer also to **A3**).
> 2. PPO is **at least 10 times more computationally expensive** than our method. Our experiment finishes in a bit more than 1 hour using H100 GPU sever, while PPO takes about a day because it requires online data sampling and learning separate reward model/value model/policy model.
> 3. **Our method is as efficient as DPO**, becuase **DPO is simply a special case of our method**. If DPO uses only pairwise responses, and InfoNCA uses 4 responses, then InfoCNA requires about 2 times the computational resource. This is reasonable because there are two times of data. Besides, using the same amount of data, our methods outperform DPO (Table 2, Line 212 of our paper).
> 4. We have compared with previous work that **uses the same amount of data with our method** (Line 208, Table 2, Table 3), so **we believe the comparison is fair**. Our results show that even previous work is given the same data magnitude, InfoNCA still outperforms them.
>
> Results with the same data:
> | Experiment  | MT-bench | Alpaca | Avg. |
> |---|---|----|---|
> | DPO × $C_4^2$ (K=4)  | 73.8  | 90.3 | 82.1 |
> | DPO × 3 (K=4) | 72.2 | 91.6   | 81.9 |
> | **DPO + PL Method** (K=4) | 74.1   | 91.3 | 82.7 |
> | **InfoNCA (K=4)**  | **76.3** | **92.4**   | **84.4** |
>
> | Model| Average among 8 tasks  |
> |---|---|
> | Mixtral-7B-SFT | 24.5 |
> | $+$ DPO  | 25.5  |
> | $+$ **NCA** | **35.7** |
> | Mixtral-8*7B-SFT | 56.3  |
> | $+$ DPO | 50.1  |
> | $+$ **NCA** | **57.9** |
>
> ***
>
> **Q3: What is the amount and percentage of reward and preference data in each dataset**
>
> **A3:**   Please refer to the first point in **A2**.
> | Dataset | Type| Response number | Used in |
> |---|---|---|----|
> | UltraFeedback| 100% Reward data  | K=4 | Table 2, Figure 5(right), Figure 6(right) |
> | UltraFeedback-clipped| 100% Reward data | K=2/3/4 | Figure 4 (left)   |
> | UltraFeedback-binarized | 100% Preference data  | K=2  | Table 2, Figure 6(left) |
> | UltraInteract | 100% Preference data  | K=2  | Table 3, Figure 5(left)               |
> Please also refer to Line 178, 193,212,220, and 226 for details.
>
> ***
>
> **Q4: Since InfoNCA and NCA are reward-based methods, why not compare them with PPO?  Table 2 should also include PPO.**
>
> **A4:** We thank the reviewer for this suggestion, and present experimental results below.
>
> | Name | Annotation Type | MT-bench | AlpacaEval |
> |---|--|--|--|
> | Mixtral-7B-sft  | SFT Data  | 6.45     | 85.20      |
> | +DPO(online) | RM Preference  | 7.14     | 88.39      |
> | **+PPO(online)** | RM Reward| 6.64     | 84.20   |
> | +InfoNCA| Reward  | **7.63** | **92.35**   |
> | +NCA  | Reward   | 7.52     | 90.31      |
>
> Initial results show that PPO underperforms DPO or InfoNCA methods, perhaps due to PPO's inherent instability or our insufficient hyperparameter tunning.
>
> **Why not previously directly compare NCA with PPO?** PPO and InfoNCA target different kinds of "reward". InfoNCA targets **REAL dataset reward** and features direct optimization. In contrast, PPO targets a **Reward Model** and requires online data sampling. We prioritize comparing with DPO-like methods since **DPO is actually our special case so we can keep exactly the same hyperparameters.**
>
> ***
>
> ## Reminder
> **QA5:**
> **We refer the reviewer to the global rebuttal posted at the top of the webpage for the rest (EZsK part 2/2) of our response.** This is due to the severe page limit.

---

> > ### Comment · Reviewer_EZsK · 2024-08-09
> >
> > Thanks for the response. Since most of my questions and concerns are addressed, I will increase my overall recommendation.

---

> ### Author Response · Authors · 2024-08-08
> **Additional Response**
>
> Dear reviewer,
>
> We copy the second part of our rebuttal response here to ease reading. In our responses, we have compared with more baselines and clarified the details of the data we are using based on your suggestions. We hope we can address your concerns and are glad to answer any further questions you might have.
>
> Thank you again for your valuable feedback!
>
> # Official Response to Reviewer EZsK (2/2)
>
> **Q5: suggest comparing it with PRO (https://arxiv.org/pdf/2306.17492).**
>
> **A5:** PRO is a quite related work of ours because it can also handle response data of arbitrary length. It is also highly related to the PL method as we have discussed in **A1**. Still, PRO is essentially a **ranking/preference-based** method,  while (Info)NCA are **reward-based** methods.
>
> **Theory comparison:**
>
> **We have added a new section as Appendix G.2 in our paper to compare with PRO theoretically:** (cannot update the manuscript during rebuttal, though)
>
> The main difference between PRO and the PL+DPO method is that
> 1. PRO has different reward formulations. It formulates $r_\theta$ as the average log-probability of a sentence $\frac{1}{\|y\|}\Sigma\log \pi_\theta(y|X)$ instead of the log-probability ratio $\log \frac{\pi_\theta(y|x)}{\mu(y|x)}$ as is required by DPO.
> 2.  Because there is no $\mu$ in reward models. PRO needs to be additionally regularized by an SFT loss in order to stay close to the pretrained LLM $\mu$.
>
> **Additional Experiments:**
> | Experiment| MT-bench |
> |----|---|
> | Mixtral-7B-sft |  64.5
> | DPO + PL Method| 74.1 |
> | PRO (SFT) | 68.9 |
> | InfoNCA  | 76.3  |
>
> Overall we find PRO (PL+SFT) method helps, but slightly underperforms the PL+DPO method. We are not confident about this conclusion because we did not faithfully replicate PRO according to its source code, but only modified our trl implementation of DPO+PL to test PRO methods. We mainly search sft_weight $\in$ [0.01, 0.05, 0.1 0.5], and keep other hyperparameters consistent with train_hh.sh of PRO repo, or the trl/DPO settings.

---

> ### Author Response · Authors · 2024-08-09
>
> We appreciate the reviewer for the prompt and positive feedback and are glad we can address the reviewer's concerns.

---

### Official Review · Reviewer_EyJu · 2024-07-19

**Soundness:** 3
**Presentation:** 3
**Contribution:** 3
**Rating:** 6
**Confidence:** 4

**Summary:**

The paper addresses a weakness of DPO: it can not deal with preference data whose number of responses is larger than 2. To extend DPO, the author proposes Noise Contrastive Estimation (NCE)-based Alignment algorithms InfoNCA. The author has shown that DPO is one of the special cases of InfoNCA. To further fix the Decreased Response Likelihood issue, the author also proposes NCA with a similar derivation. Experiments demonstrate that NCA and InfoNCA outperform baselines with reward datasets for training, with NCA showing significant performance in complex reasoning tasks like math and coding. The paper offers the community a general framework for LLM alignment from a novel perspective.

**Strengths:**

1. The paper considers the LLM alignment task from a very novel perspective of Noise Contrastive Estimation. The proposed InfoNCA and NCA methods are not only general in form (DPO can be regarded as one of the special cases), but also elegant in math (the connection of LLM alignment and InfoNCE is quite interesting). The paper brings a brand-new view to the study of LLM alignment, which can be insightful to the research community.

2. The motivation of the paper is clear, which focuses on the limitation of DPO when training with multi-response preference data. This improving direction is quite valuable for both the research and industry domains.

3. The experimental results support the effectiveness of InfoNCA and NCA, which outperform baseline methods such as DPO, IPO, and KTO.

4. The paper is clearly written, well organized, and easy to follow.

**Weaknesses:**

1. Fundamental Assumption: My major concern is about the fundamental assumption when deriving the InfoNCA and NCA methods. The author assumes that one of the scored responses is sampled from the optimal policy, which is practically impossible because most of the preference data are sampled from non-optimal LLM policy. In some extreme situations, all the generated responses can be harmful, while our ideal optimal policy should output harmless responses, leading to a contradiction with the author's essential assumption. A possible solution to this weakness might be mixing the human written (golden) response into the preference data, then applying NCE to the prompt and the golden response, combining InfoNCA with some prior work such as APO [1] and SPIN [2].

2. Lack of Human Evaluation: Although the author has conducted GPT-4 evaluation in the experiments, the real human evaluation results are not reported, which should have made the paper's claims more persuasive.

3. Lack an experiments of two-response alignment: Although the paper demonstrates better performance using multi-response preference data, it is interesting to compare InfoNCA, NCA, and DPO on a binary-response preference set.

4. Lack of baseline comparison: Although DPO cannot deal with multi-response preference, one can first train an RM with the multi-response preference, and next sample two responses with scores from the learned RM. Then DPO can be conducted on the RM-scored pairwise preference data. This can be a simple but important baseline that the author should take into consideration.

References:

[1] Adversarial Preference Optimization: Enhancing Your Alignment via RM-LLM Game, ACL 2024

[2] Self-Play Fine-Tuning Converts Weak Language Models to Strong Language Models, ICML 2024

**Questions:**

See Weaknesses.

**Limitations:**

The author has NOT addressed the limitations of the proposed method. One of the most important limitations can be that if all the scored responses are harmful, the proposed methods can never achieve the ideal alignment optimum, which could even enhance the harmful model outputs.

---

> ### Author Rebuttal · Authors · 2024-08-06
>
> # Official Response to Reviewer EyJu (1/2)
>
> **Q1: (Major Concern) Fundamental Assumption is we can sample from the optimal policy $\pi^\*$, which is practically impossible because we can only access non-optimal LLM policy $\mu$.**
>
> **A1:** We'd like to clarify that **the availability of** $\pi^\*$ **is NOT a concern of our proposed (Info)NCA algorithm, both theoretically and practically**.
>
> We are glad the reviewer is interested in this question, though, because it is a **core contribution/difficulty in our algorithm's derivation.**
>
> Take InfoNCA loss for instance (Eq.6, Line 110 in the paper):
>
> $L_\theta^\text{InfoNCA} = - E_{p(x) \prod \mu(y_i|x)} \sum_{i=1}^K [\frac{e^{r(x,y_i)/\alpha}}{Z(x)} \log \frac{e^{r_\theta(x,y_i)}}{\sum_{j=1}^K e^{r_\theta(x,y_j)}}]$
>
> Here $\mu$ is the pretrained LLM distribution. We can see the optimal policy $\pi^*$ does not show up in the loss function, which means $\pi^*$ **is NOT required**. From $E_{p(x) \prod \mu(y_i|x)}$ we can also see that, ideally all responses should be sampled from suboptimal $\mu$ instead of $\pi^*$.
>
> **Why $\pi^\*$ does not show up in the loss function?**  After all $\pi^\*$ is required/used for (Info)NCA algorithm derivation (Figure 3, Line 98 and 146, etc.):
>
> Recall that  $\pi^\*:= \mu(y|x)\frac{e^{r(x,y)/\alpha}}{Z(x)}$ is actually defined by the pretrained LLM $\mu$ and a reward function $r$. Thus **$\pi^\*$ can actually replaced by $r$ and $\mu$ in practice. This is exactly why we need dataset "rewards"**. Specifically, this is accomplished by the "rejection sampling" technique. Image we can first sample $K >> 1$ responses $y_{1:K}$ from $\mu(\cdot|x)$, score each response with $r_{1:K}$, then we can resample a single response $y$ from $y_{1:K}$ with the probabilty ratio $p_k \propto e^{r_k/\alpha}$. In this way, the distribution of $p(y) = \mu(y|x)\frac{e^{r(x,y)/\alpha}}{Z(x)}$ can be proved if K is infinitely large.
>
> This idea can be applied to elimitate the requirement of  $\pi^\*$ by importance sampling in algorithm derivation:
> $E_{\pi^*(y|x)} f(x,y) = E_{\mu(y|x)} \frac{e^{r(x,y)/\alpha}}{Z(x)}f(x,y)$
>
> so that
>
> $L_\theta^\text{InfoNCA} = - E_{p(x) \prod \pi^\*(y_i|x)} \sum_{i=1}^K \left[\log \frac{e^{r_\theta(x,y_i)}}{\sum_{j=1}^K e^{r_\theta(x,y_j)}}\right] $
>
> $= - E_{p(x) \prod \mu(y_i|x)} \sum_{i=1}^K \left[\frac{e^{r(x,y_i)/\alpha}}{Z(x)} \log \frac{e^{r_\theta(x,y_i)}}{\sum_{j=1}^K e^{r_\theta(x,y_j)}}\right]$ (proof Line 463 in Appendix)
>
>
> ***
> **Q2: In some extreme situations, all the generated responses can be harmful, while our ideal optimal policy should output harmless responses. (Contradict with Fundamental Assumption). If all the scored responses are harmful, the proposed methods can never achieve the ideal alignment optimum ...**
>
> **A2:** **We think the above problem arises fundamentally because of the problem definition of alignment  tasks instead of our proposed (Info)NCA algorithm.**
>
> Recall that most alignment problems define the desired policy distribution as:
>  $\pi^*\propto \mu(y|x)e^{r(x,y)/\alpha}$.
>
> **By definition,** we not only want $\pi^\*$ to maximize  reward $r(x,y)$, but **we want $\pi^\*$ to stay close to pretrained policy $\mu$** as well.  If all responses are harmful, then $\mu$ must be a very low-quality distributional prior for $\pi^\*$ (All responses should be sampled from $\mu$ as we have explained in **A1**). Consequently, it is natural that $\pi^*$ may struggle to get high performance because it needs to stay close to a low-quality LLM $\mu$.
>
> ***
> **Q3: Mixing the human written (golden) response into the preference data, then applying NCE to the prompt and the golden response( combining InfoNCA with some prior work such as APO [1] and SPIN [2].)**
>
> **A3:** We agree with the reviewer. Mixing golden responses into datasets would increase data quality because these responses do not come from the suboptimal policy $\mu$. Instead, they are from a presumably superior policy (human). This strategy is illuminating and would surely be very helpful if we can somehow access high-quality data beyond just a suboptimal policy $\mu$.
>
> The basic idea of APO/SPIN is to perform some GAN-like training and contrast STF/golden data with model-generated responses. Their contrastive training style can be readily combined with (Info)NCA training methods.  **This is a very promising research topic, though we suppose it is beyond the discussion scope of our current article** because we focus on the preference/reward training paradigm. **We believe this does not hurt or even support the core contribution of our method.**
>
> ***
> **Q4: Comparison with another baseline:  first train an RM with the multi-response preference, and next sample two responses with scores from the learned RM. Then DPO can be conducted on the RM-scored pairwise preference data.**
>
> **A4:** We conduct additional experiments (online DPO setting and PPO settings) and present the results below. RM is pretrained on the UltraFeedback datasets, named UltraRM.
>
> Overall, results that are trained by applying RM underperforms directly finetuned from dataset rewards. We believe this is mainly because of the learned RM is not robust enough compared with the GPT-4 annotator.
> | Name                       | Annotation Type | MT-bench | AlpacaEval |
> |----------------------------|-----------------|----------|------------|
> | Mixtral-7B-sft             | SFT Data        | 6.45     | 85.20      |
> | +KTO        | Preference      | 7.12     | 91.93      |
> | +IPO         | Preference      | 7.45     | 90.62      |
> | **+DPO(online)** | RM Preference      | 7.14     | 88.39      |
> | +PPO(online) | RM Reward          | 6.64     | 84.20      |
> | +InfoNCA        | Reward          | **7.63** | **92.35**   |
> | +NCA            | Reward          | 7.52     | 90.31      |
>
>
> ***
>
> ## Reminder
> **QA5-QA6:**
> **We refer the reviewer to the global rebuttal posted at the top of the webpage for the rest (EyJu part 2/2) of our response.** This is due to the severe page limit.

---

> ### Author Response · Authors · 2024-08-08
> **Additional Response**
>
> Dear reviewer,
>
> We copy the second part of our rebuttal response here to ease reading. In our responses, we have conducted additional experiments and compared with more baselines based on your suggestions. We hope we can address your concerns and are glad to answer any further questions you might have.
>
> Thank you again for your valuable feedback!
>
> # Official Response to Reviewer EyJu (2/2)
>
> **Q5: Lack  of two-response alignment:  compare InfoNCA, NCA, and DPO on a binary-response preference set.**
>
> **A5:** Perhaps we did not fully understand the reviewer's comment. As a matter of fact, **the whole section 5.2 aims to compare these algorithms on binary-response preference sets.** This includes Table  3, Figure 4 (left), Figure 5, and Figure 6(left).
>
> Note that in pairwise-preference settings, InfoNCA becomes exactly the DPO algorithm, so we mainly compare NCA and DPO (InfoNCA).
>
> We copy the main results below:
> **UltraFeedback (binary)**
> | Method  | MT-bench | Alpaca |
> |---|---|--|
> | InfoNCA/DPO | 73.8     | 90.7   |
> | NCA     | 73.2     | 89.9   |
>
> **UltraInteract (binary)**
>
>
> |Model|Average|Reasoning|LeetCode|HumanEval|GSMPLUS|MATH|TheoremQA|SVAMP|ASDiv|
> |---|---|---|---|---|---|---|---|---|---|
> |Mixtral-7B-SFT|24.5|60.9|3.3|28.1|28.5|5.8|7.0|26.9|35.8|
> |$+$DPO|25.5|**61.7**|2.2|**31.7**|12.1|6.4|9.8|34.1|46.1|
> |$+$**NCA**|**35.7**|60.8|**3.3**|26.8|**32.3**|**11.7**|**11.0**|**65.3**|**74.3**|
> |Mixtral-8*7B-SFT|56.3|**75.6**|16.7|61.0|57.6|40.1|25.9|85.9|87.5|
> |$+$DPO|50.1|74.9|17.2|47.6|55.8|35.3|**26.9**|67.3|75.7|
> |$+$**NCA**|**57.9**|**75.6**|**21.1**|**62.8**|**61.5**|**41.6**|**26.9**|**86.8**|**86.9**|
>
> **Q6: The real human evaluation results are not reported, which should have made the paper's claims more persuasive.**
>
> **A6:** **We have actually reported human evaluation results in the original paper** and would like to refer the reviewer to Table 2 (win vs DPO), and  Figure 7in the paper.
>
> Still, GPT-4 based evaluation results are most indicative to ourselves because datasets are annotated by GPT-4. After all, you want your training rewards and test rewards to be aligned to make sure the algorithm itself is valid given that our work is somehow more theoretical rather than product-oriented.

---

> ### Author Response · Authors · 2024-08-12
> **Looking forward to your feedback**
>
> Dear reviewer,
>
> Thank you for your valuable comments regarding our submission.  We have posted our responses and hope to address your concerns about the fundamental assumption of our method (availability of the optimal policy), as well as the comparison with several baselines such as APO and SPIN.
>
> Could you please take a look at our responses and give us some further feedback so that we can continually improve our work? Thank you so much in advance!
>
> Best regards,
>
> The Authors

---

> > ### Comment · Reviewer_EyJu · 2024-08-14
> >
> > Thank you for your elaborate response. Since the author has expanded with more comprehensive and detailed experiments, I have decided to raise my score.

---

> > > ### Author Response · Authors · 2024-08-14
> > >
> > > Thank you for your feedback. We are glad that our responses help.

---

### Author Rebuttal · Authors · 2024-08-06

# Rebuttal Summary

We would like to thank all the reviewers for their valuable comments. We are encouraged to see all reviewers recognize the theoretical contribution of our work. Reviewers EyJu, EZsK, and UYN1 highlight the importance of our work in unifying contrastive learning theories (NCE) with existing alignment methods. Reviewer EZsK, WFsu, and xAZC feel that NCA being able to prevent likelihood decline is quite meaningful. Reviewer xAZC and  EZsK think our methods being able to handle explicit reward data is a good contribution.

Concerns primarily relate to further comparisons with some related methods, and some confusion about the contrastive learning theories in our work.

Below, we summarize the main actions taken during the rebuttal:
1. We additionally conduct experiments of Plackett-Luce model, PRO method, Online DPO, and PPO, and compare them with our methods.
2. We theoretically compare our methods with APO, SPIN, SimPO, ODPO, and PRO methods.
3. We clarify some confusion or misunderstanding regarding our methods and experiments.
4. We update our presentations in paper to be more strict and clear.

Looking forward to further discussions with the reviewers!

---

|

|

|

***


# Official Response to Reviewer EyJu (2/2)
continue due to the page limit


**Q5: Lack  of two-response alignment:  compare InfoNCA, NCA, and DPO on a binary-response preference set.**

**A5:** Perhaps we did not fully understand the reviewer's comment. As a matter of fact, **the whole section 5.2 aims to compare these algorithms on binary-response preference sets.** This includes Table  3, Figure 4 (left), Figure 5, and Figure 6(left).

Note that in pairwise-preference settings, InfoNCA becomes exactly the DPO algorithm, so we mainly compare NCA and DPO (InfoNCA).

We copy the main results below:
**UltraFeedback (binary)**
| Method  | MT-bench | Alpaca |
|---|---|--|
| InfoNCA/DPO | 73.8     | 90.7   |
| NCA     | 73.2     | 89.9   |

**UltraInteract (binary)**


|Model|Average|Reasoning|LeetCode|HumanEval|GSMPLUS|MATH|TheoremQA|SVAMP|ASDiv|
|---|---|---|---|---|---|---|---|---|---|
|Mixtral-7B-SFT|24.5|60.9|3.3|28.1|28.5|5.8|7.0|26.9|35.8|
|$+$DPO|25.5|**61.7**|2.2|**31.7**|12.1|6.4|9.8|34.1|46.1|
|$+$**NCA**|**35.7**|60.8|**3.3**|26.8|**32.3**|**11.7**|**11.0**|**65.3**|**74.3**|
|Mixtral-8*7B-SFT|56.3|**75.6**|16.7|61.0|57.6|40.1|25.9|85.9|87.5|
|$+$DPO|50.1|74.9|17.2|47.6|55.8|35.3|**26.9**|67.3|75.7|
|$+$**NCA**|**57.9**|**75.6**|**21.1**|**62.8**|**61.5**|**41.6**|**26.9**|**86.8**|**86.9**|

***

**Q6: The real human evaluation results are not reported, which should have made the paper's claims more persuasive.**

**A6:** **We have reported human evaluation results in the original paper** and would like to refer the reviewer to Table 2 (win vs DPO), and  Figure 7in the paper.

Still, GPT-4 based evaluation results are most indicative to ourselves because datasets are annotated by GPT-4. After all, you want your training rewards and test rewards to be aligned to make sure the algorithm itself is valid given that our work is somehow more theoretical rather than product-oriented.

---

|

|

|

***
# Official Response to Reviewer EZsK (2/2)
continue due to the page limit

**Q5: suggest comparing it with PRO (https://arxiv.org/pdf/2306.17492).**

**A5:** PRO is a quite related work of ours because it can also handle response data of arbitrary number. It is also highly related to the PL method as we have discussed in **A1**. Still, PRO is essentially a **ranking/preference-based** method,  while (Info)NCA are **reward-based** methods.

**Theories comparison:**

**We have added a new section as Appendix G.2 in our paper to compare with PRO theoretically:** (cannot update the manuscript during rebuttal, though)

The main difference between PRO and the PL+DPO method is that
1. PRO has different reward formulations. It formulates $r_\theta$ as the average log-probability of a sentence $\frac{1}{\|y\|}\Sigma\log \pi_\theta(y|X)$ instead of the log-probability ratio $\log \frac{\pi_\theta(y|x)}{\mu(y|x)}$ as is required by DPO.
2.  Because there is no $\mu$ in reward models. PRO needs to be additionally regularized by an SFT loss in order to stay close to the pretrained LLM $\mu$.

**Additional Experiments:**
| Experiment| MT-bench |
|----|---|
| Mixtral-7B-sft |  64.5
| DPO + PL Method| 74.1 |
| PRO (SFT) | 68.9 |
| InfoNCA  | 76.3  |

Overall we find PRO (PL+SFT) method helps, but slightly underperforms the PL+DPO method. We are not confident about this conclusion because we did not faithfully replicate PRO according to its source code, but only modified our trl implementation of DPO+PL to test PRO methods. We mainly search sft_weight $\in$ [0.01, 0.05, 0.1 0.5], and keep other hyperparameters consistent with train_hh.sh of PRO repo, or the trl/DPO settings.

---

### Comment · Area_Chair_FQG2 · 2024-08-09
**Reviewers, please respond to the author rebuttals**

Dear reviewers, this is your AC!

The authors have responded to your review. Could you please take a look at the rebuttal (and respond/edit your scores if need be)?
(Thank you if you have already done so)

Thanks!

---

### Decision · Program_Chairs · 2024-09-25

**Decision:**

Accept (poster)

**Comment:**

This paper proposes a noise-contrastive estimation-based approach to alignment in which DPO can be seen as a special case. The paper is well-written, theoretically-grounded, and backed up by solid empirical experiments.  A clear accept.

(There were some questions about the claimed novelty on top of DPO, given DPO's generalization to ranked responses via the Plackett-Luce model is not only straightforward, but explicitly mentioned in the original paper's appendix. I recommend the authors to add more discussion around this point.)